# Analysis on Trade Competition and Complementarity of High-Quality Agricultural Products in Countries along the Belt and Road Initiative

**Xiao Wang [1,*], Jinming Shi [2], Jia Li [2], Yu Chen [1], Jianxu Liu [1] and Songsak Sriboonchitta [3]**

[1] School of Economics, Shandong University of Finance and Economics, Jinan 250014, China; 212101001@mail.sdufe.edu.cn (Y.C.); 20180881@sdufe.edu.cn (J.L.)
[2] School of Economics, Shandong Normal University, Jinan 250014, China; 2022010073@stu.sdnu.edu.cn (J.S.); 614004@sdnu.edu.cn (J.L.)
[3] Faculty of Economics, Chiang Mai University, Chiang Mai 50200, Thailand; songsakecon@gmail.com
[*] Correspondence: 20108594@sdufe.edu.cn

**Abstract:** The Belt and Road Initiative was proposed by China in 2013 as a response to sluggish global economic growth. With most countries along the Belt and Road being developing countries, it is crucial to strengthen trade cooperation in agricultural products. However, the current literature lacks an analysis of the competitiveness and complementarity of agricultural products in these countries. This study aims to fill this gap by showing that the Belt and Road Initiative has reduced agricultural export competitiveness and increased agricultural trade complementarity. Several factors influence the similarity and complementarity of agricultural exports in participating countries along the Belt and Road, including geographical distance, level of economic development, free trade agreements, degree of country openness, exchange rates, cultural differences, share of agricultural value added, and level of infrastructure. The detailed analysis shows that the Belt and Road Initiative has significantly improved the quality of the agricultural exports of participating countries. The results of this paper provide a theoretical basis for the high-quality development of agricultural products in participating countries along the Belt and Road.

**Keywords:** Belt and Road Initiative; export competitiveness of agricultural products; trade complementarity of agricultural products

## 1. Introduction

In 2013, Chinese government proposed the Belt and Road Initiative, and has adhered to the development concept featuring peace, development, cooperation, and win-win results [1]. China and participating countries along the route are actively engaged in industrial docking in the agricultural sector, and methods of cooperation have been enriched [2]. The Belt and Road Initiative has expanded the agricultural trade markets of the participating countries along the route, achieving mutual benefits and win-win results. Since most of the participating countries along the route are developing countries, the level of agricultural development has implications for the livelihood of the people and national security [3]. The analysis of the development trend of agricultural trade in the participating countries along the Belt and Road is of great importance for strengthening agricultural trade cooperation. At the same time, it can provide a basis for countries along the route to adjust their foreign trade policies according to their comparative advantages in agricultural products.

The literature dealing with the subject of this article focuses on two main directions. One direction focused on the impact of the Belt and Road Initiative on the trade of participating countries along the route. Wu et al. [4] found that the Belt and Road Initiative helps countries along the route to participate in global value chains. Zheng et al. [5] argue

that the Belt and Road Initiative can effectively promote value chains between China and participating countries along the route. There is also literature on the impact of the Belt and Road Initiative on the binary margins of exports, the quality of export products, and the value added of exports of participating countries along the route [6–9]. The second category is the impact of the Belt and Road Initiative on agricultural trade of participating countries along the route. Zhou and Tong [10] find large differences in agricultural export competitiveness between China and participating countries along the route. Liu et al. [11] find that agricultural trade between participating countries along the Belt and Road is complementary rather than competitive. Zhang et al. [12] find that the Belt and Road Initiative has improved the position in the global agricultural value chain, with participating countries along the route having an advantage.

The other direction is about the approach to complementarity and competition in agricultural trade. The first approach analyzes the complementarity and competitiveness of agricultural products between countries based on the Trade Combination Degree (TCD). The higher the TCD value, the closer the trade between the two countries [13]. Bouët et al. [14] argue that Africa has weaker trade integration with the rest of the world. The second approach is analyzed in terms of revealed comparative advantage (RCA). The index of revealed comparative advantage is also an important indicator of international competitiveness [15]. Zhang [16] finds that China and Brazil's agricultural exports are relatively less competitive and more complementary. Zhou et al. [10] use the RCA to suggest that the international competitiveness of agricultural products differs significantly between China and participating countries along the Belt and Road route. The third approach uses the trade specialization index (CS) and the trade consistency index (CC) to analyze the complementarity and competitiveness of agricultural products between the two countries [17]. Nurgazina et al. [18] show that although the gap between exports of agricultural products from Kazakhstan and China is decreasing, the complementarity between the exports of the two countries is gradually increasing. Some other scholars have analyzed the factors affecting complementarity and competitiveness of agricultural trade, such as geographical distance, climatic environment, population growth, level of infrastructure, size of economy, exchange rate, inflation, and free trade agreements [19–27].

Previous studies have examined the competitiveness and complementarity of agricultural trade between countries along the Belt and Road Initiative only in relation to individual countries. However, the competitiveness and complementarity of agricultural trade between countries still varies widely. For most countries along the Belt and Road Initiative, there is also a need to achieve complementarity in agriculture through mutual cooperation to promote sustainable agricultural development. This paper analyzes the comparative advantages of agricultural trade in participating countries along the Belt and Road to make recommendations for quality agricultural development. The marginal contributions of this paper are:

First, this paper is one of the few studies that examines the competitiveness and complementarity of agricultural exports of participating countries along the Belt and Road Initiative. This paper measures the competitiveness and complementarity of agricultural trade among key participating countries along the Belt and Road from 2012 to 2020 and finds that agricultural trade among participating countries along the Belt and Road is less competitive and complementary, with more room for trade cooperation.

Second, this paper examines the impact of the Belt and Road Initiative on the quality of agricultural exports of participating countries along the route and provides support for the promotion of high-quality agricultural development in participating countries along the route.

The following sections in this paper are arranged in sequence as Section 2—Materials and Methods, Section 3—Results, Section 4—Discussion, and Section 5—Conclusions.

## 2. Materials and Methods

In this paper, the indexes of revealed comparative advantage (RCA), the export similarity index (ESI), and the trade complementarity index (TCI) are used to analyze the export potential of agricultural products of countries along the Belt and Road.

The RCA index was introduced by Balassa, who utilized import and export trade data to indirectly evaluate the comparative advantage of products [15]. Istudor et al. furthered this approach by using a value chain perspective to gauge the export competitiveness of agricultural products [28]. However, this method necessitates a complete agricultural value chain, which is not present in developing countries along the Belt and Road where trade in agricultural products is concentrated in primary agricultural products. Therefore, we continue to use the RCA index to analyze the export competitiveness of agricultural products in this context.

The ESI was developed by Finger and Kreinin to assess the similarity of export products by analyzing the presence of the same product exported to the same country by two countries or regions [29]. The TCI, proposed by Grubel and Lloyd, measures the degree of correspondence between import and export products in two countries. A higher value indicates a stronger complementarity between the two countries [30].

### 2.1. Research Methods

#### 2.1.1. The Revealed Comparative Advantage (RCA) of Agricultural Products

The RCA for agricultural products refers to a value for the share of agricultural products in a given region in the country's (or region's) exports relative to that industry's share of world trade. It is used to assess the comparative trade advantage of a country's (or region's) agricultural products and to measure their international competitiveness.

$$RCA_{iK} = (X_{iK}/X_{it})/(X_{wK}/X_{wt}) \tag{1}$$

where *i* represents a country (or region) and *K* represents agricultural products (mainly including planting, forestry, animal husbandry, agricultural and sideline industries, and aquaculture). $X_{it}$ and $X_{wt}$ represent the total agricultural exports of a country (or region) and of the world, respectively; $X_{iK}$ and $X_{wK}$ represent the total exports of agricultural product *K*. Equation (1) analyzes the ratio of a share of a country's exports in global exports commodities to a share of its total exports in total global exports. If the larger the global export share of goods j of country i in export share of all products in the country, the RCA is greater than 1, indicating that a country has a "revealed" comparative advantage in production of specific goods. Specifically, if the index $RCA_{iK}$ is less than 1, it is suggested that the agricultural products in the country (or region) have comparative disadvantages, without international competitiveness. If $1 < RCA_{iK}$ index $\leq 1.25$, it is suggested that the agricultural products in the country (or region) have comparative advantages with certain international competitiveness. If $1.25 < RCA_{iK}$ Indx $\leq 2.5$, it is suggested that the agricultural products in the country (or region) have strong international competitiveness. If $RCA_{iK}$ Index $> 2.5$, it is suggested that the agricultural products in the country (or region) have extremely strong international competitiveness.

#### 2.1.2. The Export Similarity Index (ESI) of Agricultural Products

The export similarity index of agricultural products is used to measure the similarity degree of agricultural products exported by any two countries (or regions) to the global (or third country) market. The larger the index, the higher the export similarity degree, and the stronger their competitiveness.

$$ESI_{ijK} = \sum_{h \in K} \left( \frac{\frac{X_{iw}^h}{X_{iw}} + \frac{X_{jw}^h}{X_{jw}}}{2} \right) * \left( 1 - \left| \frac{\frac{X_{iw}^h}{X_{iw}} - \frac{X_{jw}^h}{X_{jw}}}{\frac{X_{iw}^h}{X_{iw}} + \frac{X_{jw}^h}{X_{jw}}} \right| \right) \tag{2}$$

where X represents exports, *h* represents HS6-bit products, and *K* represents agricultural products. $X_{iw}$ and $X_{jw}$ represent the total agricultural products exports of country (or region) i and country (or region) j to the world, respectively, and $X_{jw}^h$ represents the total agricultural products exports of country (or region) i and country (or region) j to the world, respectively. The closer the index $ESI_{ijK}$ is to 1, the higher the export similarity between the two countries or regions, and the stronger the trade competitiveness.

2.1.3. The Trade Complementarity Index (TCI) of Agricultural Products

The agricultural product trade complementarity index is calculated by the arithmetic average $TCI_{ijK}$ of the modified specialization coefficient $CS_{ijK}$ and consistency coefficient $CC_{ijK}$. The index is used to measure the compatibility of agricultural products in two countries or regions. If the $TCI_{ijK}$ of two countries (or regions) is higher, it is suggested that there is extremely high trade potential of agricultural products in the two countries (or regions).

$$CS_{ijK} = 1 - \frac{\sum_{h \in K} \left| E_{it}^h - I_{jt}^h \right|}{2} \tag{3}$$

$$CC_{ijK} = \frac{\sum_{h \in K} E_{it}^h * I_{jt}^h}{\sqrt{\sum_{h \in K} \left(E_{it}^h\right)^2 \sum_{h \in K} \left(I_{it}^h\right)^2}} \tag{4}$$

$$TCI_{ijK} = \frac{CS_{ijK} + CC_{ijK}}{2} \tag{5}$$

where $E_{it}^h$ represents the proportion of the $h^{th}$ HS6-bit agricultural products exported by country *i* (or region) in the total agricultural products exported by the country (or region) in the year t, and $I_{it}^h$ represents the proportion of the $h^{th}$ HS6-bit agricultural products imported by the country (or region) in the total agricultural products imported by the country (or region) in the year t. $CS_{ijK}$ represents the specialization coefficient of agricultural products *K*, and $CC_{ijK}$ represents the consistency coefficient of agricultural products. $CI_{ijK}$ is used to measure the trade complementarity index of countries (or regions) *i* and *j* in relation to agricultural products. The closer the value of $CI_{ijK}$ is to 1, the stronger the trade complementarity of agricultural products between the two countries (or regions).

*2.2. Data Sources and Explanations*

The trade indicator measurement data in this paper is taken from the UN COMTRADE database from 2012 to 2020. Considering that many countries are involved in the Belt and Road Initiative, this paper selects 65 representative countries and divides them into 7 regions according to their geographical location (see Table 1 and Figure 1). According to the International Convention on the Harmonized Commodity Description and Coding System (HS), chapters 1–24 and 44–46 are divided into agricultural products in this paper. To allow comparison and analysis of agricultural products in different countries (or regions), agricultural products are divided into five categories (see Table 2), namely crop production, forestry, livestock, agriculture and sideline (processing and sale of agricultural products and sideline), and aquaculture.

**Table 1.** Key countries along the Belt and Road.

| Area | Main Countries |
| --- | --- |
| China | China |
| Central Asia (5 countries) | Kazakhstan, Kyrgyzstan, Tajikistan, Uzbekistan, Turkmenistan |
| Northeast Asia (2 countries) | Mongolia, Russia |
| Southeast Asia (11 countries) | Vietnam, Laos, Cambodia, Thailand, Malaysia, Singapore, Indonesia, Brunei, Philippines, Myanmar, Timor-Leste |

**Table 1.** *Cont.*

| Area | Main Countries |
|---|---|
| South Asia (8 countries) | India, Pakistan, Bangladesh, Afghanistan, Nepal, Bhutan, Sri Lanka, Maldives |
| Central and Eastern Europe (19 countries) | Poland, Czech Republic, Slovakia, Hungary, Slovenia, Croatia, Romania, Bulgaria, Serbia, Montenegro, Macedonia, Bosnia and Herzegovina, Albania, Estonia, Lithuania, Latvia, Ukraine, Belarus, Moldova |
| South Asia, Middle East (19 countries) | Turkey, Iran, Syria, Iraq, United Arab Emirates, Saudi Arabia, Qatar, Bahrain, Kuwait, Lebanon, Oman, Yemen, Jordan, Israel, Palestine, Armenia, Georgia, Azerbaijan, Egypt |

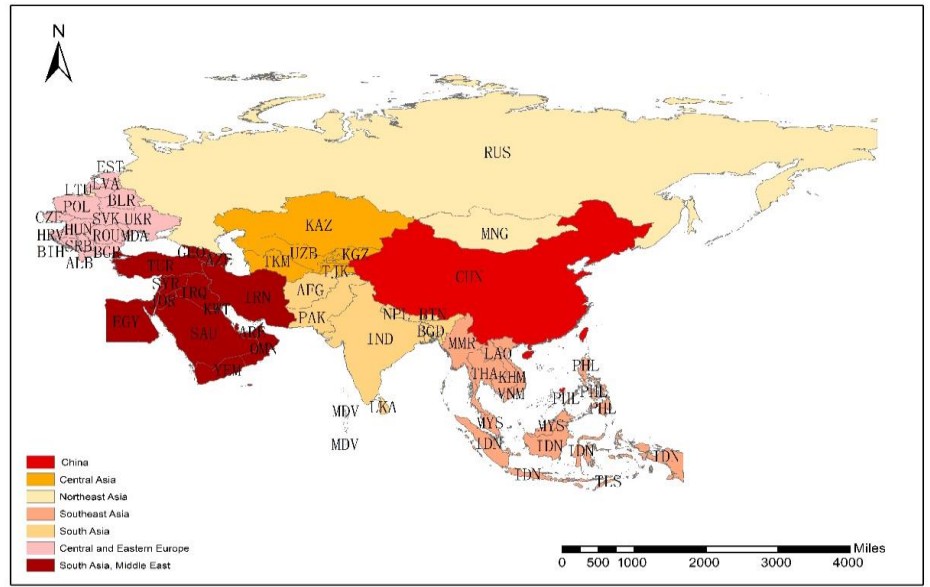

**Figure 1.** Map of countries along the Belt and Road. Notes: Author's own calculations based on ArcGIS 10.7 and the Global Administrative Unit Layers are the data produced by FAO International with funding support from organizations such as the Bill and Melinda Gates Foundation.

**Table 2.** Division of agricultural products.

| Agricultural Products Industry | HS Coding | Detailed Description |
|---|---|---|
| Planting industry | Chapters 6–14, 17–20, 24 | Planting products |
| Forest industry | Chapter 44–46 | Wood and wood products, charcoal, cork and cork products, woven goods |
| Animal husbandry | Chapters 1–2, 4–5, 16 (catalogue 1–3) | Live animals, meat, dairy products, eggs, honey and other animal products |
| Agricultural and sideline industry | Chapter 21–23 | Beverage, wine, vinegar, residues of food industry, animal feed |
| Aquaculture | Chapter 3, chapter 16 (catalogue 4) | Fish and other aquatic animals, aquatic invertebrate products |

## 3. Results

*3.1. The Competitiveness and Complementarity of the Agricultural Product Trade*

3.1.1. Comparative Advantage Analysis of Agricultural Products

In this paper, the UN COMTRADE database and Equation (1) are jointly used to calculate the RCA of agricultural products in the participating countries (or regions) along the Belt and Road Initiative (see Tables 3 and A1 for the results).

**Table 3.** RCA of agricultural products in the participating countries (or regions) along the Belt and Road Initiative.

| Type | Year | China | Central Asia | Northeast Asia | Southeast Asia | South Asia | Central and Eastern Europe | South Asia, Middle East |
|---|---|---|---|---|---|---|---|---|
| Planting industry | 2012 | 0.7771 | 1.8389 | 0.9369 | 0.9620 | 1.4327 | 1.0178 | 1.4210 |
| | 2020 | 0.8654 | 1.3903 | 0.6009 | 0.9856 | 1.4219 | 0.9896 | 1.2885 |
| Forest industry | 2012 | 1.5164 | 0.0904 | 1.6462 | 1.8218 | 0.1228 | 1.7005 | 0.3930 |
| | 2020 | 1.4415 | 0.2275 | 0.1397 | 0.9625 | 0.4305 | 1.4144 | 0.5760 |
| Animal husbandry | 2012 | 0.6577 | 0.0994 | 0.5354 | 0.1872 | 0.4994 | 1.1678 | 0.7601 |
| | 2020 | 0.4919 | 0.6126 | 3.3771 | 0.5333 | 0.4671 | 1.0241 | 0.9942 |
| Agricultural and sideline industry | 2012 | 0.8648 | 0.2442 | 0.1726 | 0.7056 | 0.5891 | 0.7312 | 0.6222 |
| | 2020 | 0.7987 | 0.6410 | 0.9169 | 1.2085 | 0.5039 | 0.9799 | 0.7127 |
| Aquaculture | 2012 | 1.6447 | 0.2648 | 3.1705 | 2.2629 | 1.0700 | 0.4787 | 0.4215 |
| | 2020 | 1.6426 | 0.2804 | 1.1845 | 1.4771 | 1.1114 | 0.6921 | 0.4434 |

Notes: Please see Table A1 for details.

China has revealed comparative advantages in forestry and aquaculture. As shown in Table A1, the average value of the RCA of the forest industry and aquaculture industry in China is 1.4865 and 1.6040, respectively, both of which are greater than 1.25, indicating that these two industries have strong international competitiveness. For planting, animal husbandry, and agricultural and sideline industries, their RCA are increasing year-by-year although they do not have the advantages, indicating that China has expanded the opening up of agricultural products year-by-year after the Chinese government proposed the Belt and Road Initiative, and effectively utilized relevant markets and resources at home and abroad to enhance the international competitiveness of related agricultural product industries.

Five countries in Central Asia show revealed comparative advantages in the planting industry. Due to the relatively high cultivated land area in the five countries, their planting industry has also developed rapidly. As shown in Table A1, the RCA of the planting industry in these five countries is greater than 1.25, among which, Uzbekistan has the highest index, with an average value of 1.8404, suggesting that the five countries have an RCA in the planting industry of cash crops such as grains, fruits, vegetables, and oil crops. For such phenomena, we noted that after joining the Belt and Road Initiative, the RCA of agricultural products in the five countries grows steadily, suggesting that the five countries can effectively contact the global market with the help of the open platform of the initiative, enhance the opening up of all countries, improve the comparative advantages of different agricultural products, and increase their international competitiveness.

The Northeast Asian countries have an RCA in forestry and aquaculture. Because the animal husbandry in Mongolia and forest industry in Russia have a relatively large RCA index, if the data of the two countries are calculated uniformly, that will cause obvious fluctuation of the overall RCA, and thus this paper calculates the index of Russia and of Mongolia separately. From observation of the respective comparative advantage index of Russia and of Mongolia, it is suggested that Russia's forest industry has an RCA, with an average value of 2.0924 (greater than the standard of 1.25), indicating that Russia's forest industry has a strong international competitiveness. At the same time, the RCA of Russian aquaculture is more than 1.25, also indicating that Russian aquaculture has strong competitiveness in the world. Furthermore, since joining the Belt and Road Initiative, the RCA of other agricultural products in Russia has also improved to varying degrees, indicating that the international competitiveness of overall agricultural products has been improved. The average value of the RCA of animal husbandry in Mongolia is 4.8819, suggesting there is a strong international competitiveness of animal husbandry, which profits from the unique geographical advantages of Mongolia, which is rich in land resources and natural farms, and has great development potential of animal husbandry.

The Southeast Asian countries show an RCA in forestry and aquaculture. As shown in Table A1, the RCA of the forest industry and aquaculture industry in Southeast Asia is greater than 1.25, among which, Laos' forest industry has the highest revealed comparative index with 2.7181 of average value; and Vietnam's aquaculture has the highest RCA, with an average value of 2.0654, indicating that the forest industry and aquaculture industry in Southeast Asia have strong competitiveness in the world. In addition to a unique geographical location, the tropical monsoon climate and tropical rain forest climate in Southeast Asia are very suitable for the growth of trees and plants, so it is rich in aquaculture resources, thus the forest industry and aquaculture industry in Southeast Asia demonstrate an RCA. Generally speaking, since Southeast Asian countries have joined the Belt and Road Initiative in 2016, the RCA of their agricultural products that used to be at an export inferior position has also steadily increased, indicating that the agricultural products in Southeast Asian countries are also developing a higher quality.

South Asian countries have an RCA in planting and aquaculture. On the one hand, South Asia is hot and rainy, with flat terrain and vast cultivated land, which is beneficial to the development of planting. On the other hand, there is a long coastline and intensive inland rivers in South Asia, so its' aquaculture is very rich. As shown in Table A1 the RCA of planting and aquaculture in South Asia is greater than 1.25, indicating that planting and aquaculture have strong international competitiveness, among which, Afghanistan is the country with the largest RCA in planting with an average value of 1.8903, and Maldives is the country with the largest RCA in aquaculture, with an average value of 7.1723.

Central and Eastern European countries showed competitive advantages in forestry and aquaculture. First of all, the RCA of the forest industry is greater than 1.25, among which, Slovenia is the country with largest advantage in the forest industry with an average value of 2.4445. On the whole, Central and Eastern European countries show strong international competitiveness in the forest industry, which benefits from the perfect forestry management mechanism and forestry infrastructure in these countries. After 2015 (Central and Eastern European countries signed the Belt and Road Initiative agreement with each other), the RCA of forest industry in Central and Eastern European countries increased steadily, indicating that the international competitiveness of agricultural products has been further enhanced since these countries participated in the Belt and Road Initiative. Second, for planting and animal husbandry, although the RCA is less than 1.25, the average value is greater than 1, which indicates that planting and animal husbandry in Central and Eastern European countries have certain comparative advantages. Finally, for agricultural and sideline industries and aquaculture with export disadvantages, the RCA has risen steadily since 2015, indicating that the international competitiveness of agricultural products has been further enhanced since Central and Eastern European countries participated in the initiative.

Countries in Southwest Asia and the Middle East have an RCA in the planting industry. As shown in Table A1, the RCA of the planting industry in these countries is greater than 1.25, which indicates that their planting industries have strong international competitiveness, among which, the average value of the RCA of the planting industry in Iraq is up to 1.8621, which is the country with the largest RCA of a planting industry in Southwest Asia and the Middle Eastern countries. However, the forest industries, animal husbandry, agricultural and sideline industries, and aquaculture do not have international competitiveness, which is mainly because of the climate environment and geographical location of these countries.

Through a horizontal comparison, the RCA of the planting industry of five Central Asian countries, the animal husbandry of Mongolian, the forest industry of Russia, and the aquaculture of Southeast Asia are all larger than those of the other countries, so these industries have strong international competitiveness [10]. Otherwise, agricultural and sideline industry for these seven countries or regions does not have international competitiveness.

### 3.1.2. Competitive Analysis of the Agricultural Product Trade

In this paper, we calculate the competitive index of the agricultural product trade of the main participating countries (or regions) along the Belt and Road Initiative from 2012 to 2020 using an export similarity index formula (Equation (2)). Only the indexes in 2013, 2016, and 2019 are reported in this paper for easy presentation of results (see Tables 4 and A2).

**Table 4.** Export similarity of agricultural products in the participating countries (or regions) along the Belt and Road Initiative.

| Year | State | State | Planting Industry | Forest Industry | Animal Husbandry | Agricultural and Sideline Industry | Aquaculture |
|------|-------|-------|------------------|-----------------|------------------|-----------------------------------|-------------|
| 2013 | China | 5 Central Asian countries | 0.4142 | 0.0083 | 0.0210 | 0.0409 | 0.0588 |
| | | Mongolia | 0.3907 | 0.1092 | 0.0402 | 0.0903 | 0.0842 |
| | | 11 Southeast Asian countries | 0.3960 | 0.1328 | 0.0266 | 0.0596 | 0.1234 |
| | | 8 South Asian countries | 0.3243 | 0.0360 | 0.0283 | 0.0561 | 0.0985 |
| | | 19 Central and Eastern European countries | 0.3649 | 0.1369 | 0.0544 | 0.0849 | 0.0539 |
| | | Southwest Asia, 19 Middle Eastern countries | 0.3884 | 0.0281 | 0.0509 | 0.0723 | 0.0508 |
| 2016 | China | 5 Central Asian countries | 0.4258 | 0.0093 | 0.0318 | 0.0313 | 0.0556 |
| | | Mongolia | 0.4258 | 0.1071 | 0.0355 | 0.0629 | 0.0815 |
| | | 11 Southeast Asian countries | 0.3956 | 0.1357 | 0.0272 | 0.0729 | 0.1246 |
| | | 8 South Asian countries | 0.3569 | 0.0358 | 0.0216 | 0.0604 | 0.1035 |
| | | 19 Central and Eastern European countries | 0.3785 | 0.1373 | 0.0481 | 0.0918 | 0.0557 |
| | | Southwest Asia, 19 Middle Eastern countries | 0.4019 | 0.0254 | 0.0440 | 0.0837 | 0.0510 |
| 2019 | China | 5 Central Asian countries | 0.4348 | 0.0118 | 0.0345 | 0.0358 | 0.0162 |
| | | Mongolia, Russia | 0.2810 | 0.0977 | 0.0463 | 0.0611 | 0.1899 |
| | | 11 Southeast Asian countries | 0.4011 | 0.1209 | 0.0352 | 0.0735 | 0.1177 |
| | | 8 South Asian countries | 0.3696 | 0.0281 | 0.0236 | 0.0713 | 0.1124 |
| | | 19 Central and Eastern European countries | 0.3693 | 0.1303 | 0.0563 | 0.0959 | 0.0600 |
| | | Southwest Asia, 19 Middle Eastern countries | 0.4021 | 0.0168 | 0.0487 | 0.0742 | 0.0699 |

Notes: Please see Table A2 for details.

We present a competitive analysis of the agricultural product trade of China. Table A2 suggests that there is a relatively high export similarity of the planting industry of China and five Central Asian countries, with a value of 0.4, showing that the trade competitiveness is not high. Meanwhile, the export similarity of the forest industry and animal husbandry in China and 19 Central and Eastern European countries, of the agricultural and sideline industry in Russia and Mongolia, and of aquaculture in 11 Southeast Asian countries are relatively high. There is an export similarity index of less than 0.3, indicating that the trade competitiveness of China and other countries is not high and stays in a relatively stable state, so there is a larger cooperation space.

We also offer a competitive analysis of the agricultural product trade of Mongolia and Russia. Table A2 suggests that Mongolia and 19 Southwest Asian and Middle Eastern

countries have a high export similarity in the planting industry, with a value of 0.5, indicating that they are highly competitive in planting exports. Meanwhile, Mongolia, Russia, and China have a relatively high export similarity in the forest industry, the agricultural and sideline industry, and aquaculture, and five Central Asian countries have a relatively high export similarity in animal husbandry, with a value of less than 0.3, suggesting that the trade competitiveness of Mongolia, Russia, and other countries is relatively high. For most of agricultural products, since Russia and Mongolia have signed the Belt and Road agreement with China, the export similarity index of agricultural products in Mongolia, Russia, and other countries or regions has declined, so their trade cooperation potential is very high.

We present a competitive analysis of the agricultural product trade of five Central Asian countries. Table A2 suggests that the five Central Asian countries and China have a relatively high similarity in their planting industries, with an index of 0.4, indicating that the five countries do not have very high trade competitiveness in the planting industry; the five Central Asian countries and 11 Southeast Asian countries have a relatively high export similarity in animal husbandry, the same as Southwest Asia and 19 Middle Eastern countries in animal husbandry, as well as 19 Central and Eastern European countries in the agricultural and sideline industry. There is an index of less than 0.3, indicating that the trade competitiveness between the five Central Asian countries and other countries is not very high, and for most agricultural products, the export similarity index of agricultural products in the five Central Asian countries or other countries or regions has declined since the five countries signed the Belt and Road agreement.

We offer a competitive analysis of the agricultural product trade of 11 Southeast Asian countries. Table A2 suggests that 11 South Asian countries have a relatively high export similarity with Russia and Mongolia in the planting industry, with an index of 0.5, suggesting that the 11 countries and Russia and Mongolia have high trade competitiveness in the planting industry. Meanwhile, the 11 Southeast Asian countries have a relatively low export similarity with the forest industry and aquaculture of China, and with the forest industry and aquaculture of eight South Asian countries, as well as with aquaculture and the agricultural and sideline industry of 19 Central and Eastern European countries, with a similarity index of less than 0.3. This indicates that there is not very high competitiveness between the 11 countries and other countries. For most of agricultural products, the export similarity index of agricultural products in the 11 countries has declined compared with other countries and regions, indicating that the trade complementarity among these countries is continuously increasing and their cooperation spaces are expanding.

We offer a competitive analysis of the agricultural product trade of eight South Asian countries. Table A2 suggests that eight South Asian countries have a relatively high export similarity with Russia and Mongolia in the planting industry, with an index of 0.5, suggesting that there is a relatively high trade competitiveness between the eight countries and Russia and Mongolia in the planting industry. Additionally, the eight countries have a relatively high export similarity in the forest industry and aquaculture with China, and in animal husbandry and agricultural and sideline industry with 19 Central and Eastern European countries, with an index of less than 0.3. This indicates that there is not a very high competitiveness between the 11 Southeast Asian countries and these countries, and there is a relatively high trade development space among these countries.

We offer a competitive analysis of the agricultural product trade of 19 Central and Eastern European countries. Table A2 suggests that eight Central and Eastern European countries have a relatively high export similarity with Russia and Mongolia in the planting industry, with an index of 0.5, suggesting that there is a relatively high trade competitiveness between eight South Asian countries and Russia and Mongolia in the planting industry. Meanwhile, the 19 Central and Eastern European countries have a relatively high export similarity with China in the forest industry and aquaculture, and with Southwest Asia and 19 Middle Eastern countries in animal husbandry and the agricultural and sideline industry, and with five Central Asian countries in agricultural and sideline industry. This comparison

yields an index of less than 0.3, indicating that there is not a very high competitiveness between the 19 countries and these countries.

We present a competitive analysis of the agricultural product trade of Southwest Asia and 19 Middle Eastern countries. Table A2 suggests that Central and Southwest Asia and 19 Middle Eastern countries have a relatively high export similarity with Russia and Mongolia in the planting industry, with an index of 0.5, suggesting that there is a relatively high trade competitiveness between the eight South Asian countries and Russia and Mongolia in the planting industry. Meanwhile, Southwest Asia and 19 Middle Eastern countries have a relatively low export similarity with China in the forest industry and aquaculture, and with 19 Central and Eastern European countries in animal husbandry, and with five Central Asian countries in the agricultural and sideline industry. This comparison yields an index of less than 0.3, indicating that there is not very high competitiveness between the Southwest Asia, 19 Middle Eastern countries, and other countries.

Although the export similarity index of each country and region in the planting industry, with a value of 0.4, indicating the planting industry of each country and region is competitive to certain extent, there is cooperation space between the countries. However, with regard to the export similarity index of other agricultural products in a country or region, the similarity index is low, with a value of less than 0.1, indicating that the trade complementarity of other agricultural products in the country or the region is relatively high. On the whole, the export similarity of agricultural products in different countries or regions has declined gradually since 2013, suggesting that the export similarity of agricultural products in different countries or regions are effectively reduced after they joined the Belt and Road Initiative.

3.1.3. Trade Complementarity Analysis of the Agricultural Product Trade

In this paper, we calculate the trade complementarity analysis of the agricultural product trade of the participating countries (or regions) along the Belt and Road Initiative from 2012 to 2020 using the arithmetic average (Formula (5)) of the specialization coefficient (Formula (3)) and the consistency coefficient (Formula (4)). Similarly, only the trade complementarity index of agricultural products in 2021 and 2020 are reported in this paper for easy presentation of results (as shown in Tables 5 and A3).

We offer a trade complementarity analysis of agricultural products in China. China has a relatively high trade complementarity of agricultural products with other countries or regions, with a trade complementarity index of 0.5 and above, indicating that China has a strong bilateral trade complementarity of agricultural products with other countries or regions. In 2020, the trade complementarity index of agricultural products in China with other countries was higher than that in 2012, indicating that participation in the Belt and Road Initiative is conducive to promoting the growth of the agricultural product trade between China and other countries or regions.

We present a trade complementarity analysis of agricultural products in five Central Asian countries. Table A3 suggests that the five Central Asian countries have a relatively high complementarity with 19 Middle Eastern countries and Southwest Asia in the planting industry, with the eight South Asian countries in the forest industry, with the 19 Central and Eastern European countries in animal husbandry, with the 11 Southeast Asian countries in the agricultural and sideline industry, and with the 19 Central and Eastern European countries in aquaculture. Although the trade complementarity index between the five Central Asian countries and some countries or regions declined in 2020 it still remained at 0.5, indicating that there is a relatively high level of trade complementarity.

We present a trade complementarity analysis of agricultural products in Mongolia and Russia. Table A3 suggests that Russia and Mongolia have a relatively high complementarity with 11 Southeast Asian countries in the planting industry, with the eight South Asian countries in the forest industry, animal husbandry, and aquaculture, and with the 19 Central and Eastern European countries in the agricultural and sideline industry. Although the trade complementarity index between Mongolia and Russia and other countries or regions

declined in 2020, their index raised with respect to agricultural products in most countries or regions, which also indicates that participation in the Belt and Road Initiative enhanced the trade and cooperation between Russia and Mongolia and other countries or regions in agricultural products to achieve high-quality development.

**Table 5.** Trade complementarity index of agricultural products in the participating countries (or regions) along the Belt and Road Initiative.

| Year | State | State | Planting Industry | Forest Industry | Animal Husbandry | Agricultural and Sideline Industry | Aquaculture |
|------|-------|-------|-------------------|-----------------|------------------|------------------------------------|-------------|
| 2012 | China | 5 Central Asian countries | 0.1976 | 0.4773 | 0.5833 | 0.5785 | 0.5657 |
| | | Mongolia, Russia | 0.3000 | 0.5482 | 0.5977 | 0.9998 | 0.9086 |
| | | 11 Southeast Asian countries | 0.5204 | 0.9928 | 0.8755 | 0.6722 | 0.9999 |
| | | 8 South Asian countries | 0.5011 | 0.8954 | 0.9996 | 0.5069 | 0.9978 |
| | | 19 Central and Eastern European countries | 0.3660 | 0.7600 | 0.6450 | 0.6291 | 0.9887 |
| | | Southwest Asia, 19 Middle Eastern countries | 0.5034 | 0.5370 | 0.8744 | 0.6460 | 0.6188 |
| 2020 | China | 5 Central Asian countries | 0.4344 | 0.5069 | 0.5281 | 0.8479 | 0.5157 |
| | | Mongolia, Russia | 0.9560 | 0.6350 | 0.5914 | 0.7810 | 0.7575 |
| | | 11 Southeast Asian countries | 0.5905 | 0.7846 | 0.5303 | 0.7584 | 0.7765 |
| | | 8 South Asian countries | 0.3591 | 0.9967 | 0.4960 | 0.7529 | 0.5064 |
| | | 19 Central and Eastern European countries | 0.5306 | 0.6208 | 0.5595 | 0.8638 | 0.7525 |
| | | Southwest Asia, 19 Middle Eastern countries | 0.5788 | 0.5191 | 0.7019 | 0.5011 | 0.5124 |

Notes: Please see Table A3 for details.

We offer a complementarity analysis of the agricultural product trade of 11 Southeast Asian countries. Table A3 suggests that the 11 Southeast countries have a relatively high complementarity with the eight South Asian countries in the planting industry and aquaculture, with Russia and Mongolia in the forest industry, and with the five Central Asian countries in animal husbandry. Both the trade complementarity index of the forest industry and aquaculture were improved, indicating that participation into the Belt and Road Initiative also strengthens the trade between the 11 Southeast Asian countries and other countries or regions in agricultural products and is conducive to the role of the industries with an RCA in the 11 Southeast Asian countries.

We present a trade complementarity analysis of agricultural products in eight South Asian countries. Table A3 suggests that the eight South Asian countries have a relatively high complementarity with 19 Middle Eastern countries and Southwest Asia in the planting industry, with Mongolia and Russia in the forest industry, with the five Central Asian countries in the agricultural and sideline industry and in animal husbandry, and with the 11 Southeast Asian countries in aquaculture. All of the trade complementarity indexes of the forest industry, animal husbandry, and agricultural and sideline industry were raised, indicating that participation in the Belt and Road Initiative enhances the trade between the eight South Asian countries and other countries or regions in agricultural products and compensates for the inferior industries of these eight countries so as to achieve high-quality development.

We present a trade complementarity analysis of agricultural products in 19 Central and Eastern European countries. Table A3 suggests that the 19 Central and Eastern Eu-

ropean countries have an extremely high trade complementarity in the forest industry with the eight South Asian countries or the 11 Southeast Asian countries, with the five Central Asian countries in the forest industry, with Southwest Asia and 19 Middle Eastern countries in animal husbandry, with Russia and Mongolia in animal husbandry, and with the five Central Asian countries in aquaculture. Although the trade complementarity index of agricultural products in these 19 countries declined in 2020, it remained at 0.5, indicating that these countries still have a relatively high level of complementarity.

We offer a trade complementarity analysis of the agricultural product trade of Southwest Asia and 19 Middle Eastern countries. According to Table A3, South Asia and 19 Middle Eastern countries have an extremely high trade complementarity in the forest industry with the eight South Asian countries, with Russia and Mongolia in the forest industry agricultural and sideline industry, with the five Central Asian countries in animal husbandry, and in aquaculture with the eight South Asian countries. Although the trade complementarity index between Southwest Asia and 19 Middle Eastern countries and other countries or regions declined in 2020, the development trend of the 19 Central and Eastern European countries remained at 0.5, indicating that there is a relatively large trade complementarity level for these countries.

Generally speaking, the complementarity of agricultural products in the participating countries along the Belt and Road Initiative is relatively high, which shows the strong trade complementarity of these countries. At the same time, from perspective of the trade complementarity index of the participating countries along the Belt and Road Initiative, on the one hand, countries can lend advantages to the industries with an RCA in each country. On the other hand, it can better compensate for the relatively inferior industries of each country and increase the space for trade cooperation to achieve high-quality development.

### 3.2. The Influence Factors of Competitiveness and Complementarity of Agricultural Products

The previous analysis regarding the RCA, export similarity index, and trade complementarity index mainly center on China and six other regions. Next, we further study the influence factors of export similarity and trade complementarity of HS6-bit agricultural products in 65 participating countries along the Belt and Road Initiative in this paper. Specifically, the control variables selected in this paper are the distance between the two countries, the development level differences between the two countries, free trade agreements between the two countries, open-up differences between the two countries, value added differences, culture differences, exchange rate fluctuation level of the two countries, and infrastructure level differences.

The export similarity index of agricultural products (ESI) is computed as the arithmetic mean of the specialization coefficient and the complementarity coefficient of agricultural products. The trade complementarity index of agricultural products (TCL) is determined as the product of revealed export comparative advantage and revealed import comparative advantage of agricultural products in each country. Distance (DIS) is defined as the population-weighted distance between countries. Differences in development level (DPG) are calculated as the ratio of GDP per person between countries minus 1. The Free Trade Agreement (FTA) is a binary variable, which takes the value of 1 if the two countries sign a free trade agreement and 0 otherwise. Open-up differences (DOP) are computed as the ratio of the open-up differences between the two countries minus 1. Language (LAN) is also a binary variable, which takes the value of 1 if the two countries have the same official language and 0 otherwise. Difference in the proportion of value-added agriculture (DVA) is defined as the ratio of the proportion of agricultural added value in the GDP minus 1. Exchange rate fluctuation (EXR) is the exchange rate fluctuation in two countries. Differences in infrastructure level (DIN) is determined using the composite control method based on indicators such as fixed broadband, air cargo volume, internet penetration rate, wharf water discharge, per capita fixed telephone amount, per capita mobile telephone amount, and railway freight volume.

The data on the export similarity index of agricultural products, trade complementarity index of agricultural products, distance, and language were obtained from the UN-COMTRADE database. The data on differences in development level, open-up differences, difference in the proportion of agricultural added value, and differences in infrastructure level were obtained from the World Bank database. The free trade agreement information was obtained from the WTO database and the exchange rate fluctuation information was obtained from the OECD database. The specific calculation methods and data resources are shown in Table 6. Table 6 reports the HS6 quartile agricultural export competitiveness and trade complementarity index among the 65 countries from 2012 to 2020. These indexes were utilized as explanatory variables in the gravity model. As there may be some differences in the export of agricultural products among different countries, the values are relatively small. All other control variables (except dummy variables) are logarithmized.

**Table 6.** Descriptive analysis.

| Variable | Obs | Mean | Std. Dev. | Min | Max |
|----------|-----|------|-----------|-----|-----|
| ESI | 17049593 | −3.9524 | 3.8392 | −14.6835 | 3.4589 |
| TCL | 5,609,330 | −11.1765 | 3.0863 | −19.572 | −4.9170 |
| DIS | 17049593 | 8.0369 | 0.8707 | 5.1517 | 9.2415 |
| DPG | 17049593 | −0.1076 | 1.3084 | −4.0026 | 3.3739 |
| FTA | 17049593 | 0.2741 | 0.4461 | 0 | 1 |
| DOP | 17049593 | 0.4502 | 0.3472 | 0 | 1.6697 |
| LAN | 17049593 | 0.0448 | 0.2067 | 0 | 1 |
| DVA | 16397459 | 1.5668 | 1.1636 | −2.2652 | 3.3618 |
| EXR | 17049593 | 0.0616 | 4.2242 | −9.8209 | 9.7212 |
| DIN | 17049593 | −0.9477 | 1.2113 | −4.8928 | 2.2734 |

Note: Variables above are logarithmic except for dummy variables such as FTA and LAN.

### 3.2.1. The Influence Factors of Export Competitiveness of Agricultural Products

The export similarity of agricultural products, namely the export competitiveness index of agricultural products, is used to measure the similarity of exports of agricultural products structure of two countries to the same single target market (or country). This paper analyzes the influence factors of export competitiveness of agricultural products in the participating countries along the Belt and Road Initiative based on the traditional gravity model, and the relevant results are shown in Table 7. First, the geographical distance refers to the trade costs, the further the distance, the lower the possibility of two countries exporting the agricultural products to the same market, and the lower the possibility of the competitiveness of their agricultural products [31].The result suggests that the geographical distance is a significant negative value at 1% level, that is, the geographical distance significantly inhibits the competitiveness of agricultural products, and the distance causes the greatest impact on aquaculture. Secondly, the differences in development level are a positive indicator, the larger the value, the larger the differences in economic development level. Table 7 suggests that the difference in development level is significantly negative. Because the participating countries along the Belt and Road are mainly developing countries with similar economic development levels, the trade possibility of these countries is larger, and the competitiveness of agricultural products is stronger in terms of similarity demand theory. Third, free trade agreements have a significantly positive value at a level of 1%. For most agricultural products, the demand elasticity is relatively large, and free trade agreements can intensify the competition of agricultural products by reducing non-tariff barriers such as price supports for agricultural products and price subsidies [32]. Similarly, the open-up difference is a significantly negative value, which is a positive indicator. The larger the value, the larger the differences, and the smaller the possibility of exporting the agricultural products to the same market, the smaller the possibility of competition. Fourth, the common language coefficient is a negative, but not significant, value. Theoretically, the same language means the same culture, which is helpful for performing technical communication and exchange of production of agricultural products and reducing the

blind export competition [33]. At the same time, the difference coefficient of the proportion of agricultural added value has a significantly negative value at a level of 1%, which is a positive indicator. The larger its value, the greater the difference in the proportion. The greater the difference in the proportion, the larger the industrialization process difference between the two countries, that is, there are obvious differences in people's demand for different products to reduce the competitiveness of agricultural products. Fifth, the exchange rate coefficient has a significantly positive value at a level of 1%. In this paper, the exchange rate of country j to country i is used. A higher rate value indicates the currency appreciation of country i, and a lower value indicates the currency devaluation of the country. The appreciation of the exchange rate of country i significantly reduces its export scale and competitiveness of agricultural products. Sixth, the difference coefficient of the infrastructure level is a significantly negative value at a level of 1%. The larger the difference value, the larger the differences between the two countries. Good infrastructure level is conducive to the communication of production and sale information of agricultural products to reduce information asymmetry, and conducive to transportation of agricultural products to reduce the logistics costs so as to decrease their competitiveness and realize high-quality development [34].

**Table 7.** Empirical analysis results of influence factors of export competitiveness of agricultural products in the participating countries (or regions) along the Belt and Road Initiative.

| | Agriculture | Planting Industry | Forest Industry | Animal Husbandry | Agricultural and Sideline Industry | Aquaculture |
|---|---|---|---|---|---|---|
| | (1) | (2) | (3) | (4) | (5) | (6) |
| VARIABLES | ESI | ESI | ESI | ESI | ESI | ESI |
| DIS | −0.1741 *** | −0.1469 *** | −0.1952 *** | −0.1681 *** | −0.1042 *** | −0.2530 *** |
| | (0.0081) | (0.0099) | (0.0256) | (0.0196) | (0.0192) | (0.0230) |
| DPG | −0.0167 *** | −0.0205 *** | −0.0350 *** | −0.0209 ** | −0.0178 ** | 0.0007 |
| | (0.0025) | (0.0027) | (0.0062) | (0.0080) | (0.0072) | (0.0070) |
| FTA | 0.0757 *** | 0.0735 *** | 0.0487 ** | 0.0699 *** | 0.0620 ** | 0.1532 *** |
| | (0.0088) | (0.0102) | (0.0242) | (0.0229) | (0.0242) | (0.0284) |
| DOP | −0.0918 *** | −0.0345 *** | −0.1126 *** | −0.1553 *** | −0.1826 *** | −0.1181 *** |
| | (0.0111) | (0.0131) | (0.0312) | (0.0320) | (0.0409) | (0.0260) |
| LAN | −0.0148 | −0.0269 | 0.0125 | 0.1115 ** | 0.0694 ** | −0.1548 *** |
| | (0.0141) | (0.0169) | (0.0372) | (0.0459) | (0.0344) | (0.0410) |
| DVA | −0.0093 *** | −0.0032 | −0.0045 | −0.0522 *** | −0.0144 ** | −0.0140 * |
| | (0.0024) | (0.0026) | (0.0051) | (0.0095) | (0.0064) | (0.0073) |
| EXR | 0.0451 *** | 0.0549 *** | 0.0559 | 0.0415 | 0.0087 | −0.0018 |
| | (0.0112) | (0.0141) | (0.0353) | (0.0336) | (0.0421) | (0.0339) |
| DIN | −0.0509 *** | −0.0448 *** | −0.0062 | −0.1135 *** | −0.0372 * | −0.0742 *** |
| | (0.0072) | (0.0089) | (0.0152) | (0.0293) | (0.0204) | (0.0185) |
| Constant | −9.6690 *** | −9.9536 *** | −9.3923 *** | −9.6113 *** | −9.3642 *** | −9.7130 *** |
| | (0.0674) | (0.0813) | (0.2116) | (0.1697) | (0.1663) | (0.1966) |
| Fixed effect of year | Yes | Yes | Yes | Yes | Yes | Yes |
| Fixed effect of country i | Yes | Yes | Yes | Yes | Yes | Yes |
| Fixed effect of country j | Yes | Yes | Yes | Yes | Yes | Yes |
| Fixed effect of products | Yes | Yes | Yes | Yes | Yes | Yes |
| Observations | 5,372,877 | 3,087,860 | 599,241 | 626,866 | 511,809 | 547,101 |
| R-squared | 0.344 | 0.357 | 0.424 | 0.347 | 0.425 | 0.383 |

Note: *** represents significant value at a 1% level, ** represents significant value at a 5% level and * represents significant value at a 1% level. This paper empirically analyzes the standard error clustered to the level of HS6-bit agricultural products (the same below).

### 3.2.2. The Influence Factors of Export Complementarity of Agricultural Products

The complementarity calculation method of agricultural products fails to calculate the trade complementarity index of HS6-bit agricultural products correctly, and thus this paper learns from Yu's [35] method to calculate the complementarity index of agricultural products and investigates its influence factors of the 65 participating countries along the Belt and Road Initiative. The regression results are shown in Table 8.

$$TCI_{ijh} = RCA^h_{Xi} \times RCA^h_{Mj} \qquad (6)$$

where $TCI_{ijh}$ represents the trade complementarity index of agricultural products $h$ of countries $i$ and $j$. The greater the value, the stronger the trade complementarity of agricultural product $h$. $RCA^h_{Xi} = X^h_i/X_i$ represents the revealed export comparative advantage index of agricultural product $h$ of country $i$, $X^h_i$ represents the amount of export of agricultural product h of country $i$, $X_i$ represents the total amount of export of agricultural products in country $i$. $RCA^h_{Mj} = M^h_j/M_j$ represents the revealed import comparative advantage index of agricultural product h of country $j$, $M^h_j$ represents the amount of export of agricultural product $h$ of country $j$, and $M_j$ represents the total amount of import of agricultural products in country $j$.

**Table 8.** Empirical analysis results of influence factors of trade complementarity of agricultural products in the participating countries (or regions) along the Belt and Road Initiative.

| VARIABLES | Agriculture (1) | Planting Industry (2) | Forest Industry (3) | Animal Husbandry (4) | Agricultural and Sideline Industry (5) | Aquaculture (6) |
|---|---|---|---|---|---|---|
| | TCI | TCI | TCI | TCI | TCI | TCI |
| DIS | −0.0448 *** | −0.0334 *** | −0.0219 * | −0.0936 *** | −0.0203 ** | −0.0617 *** |
| | (0.0048) | (0.0056) | (0.0125) | (0.0163) | (0.0097) | (0.0120) |
| DPG | 0.0095 *** | −0.0025 | 0.0222 *** | 0.0250 *** | 0.0071 | 0.0269 *** |
| | (0.0022) | (0.0024) | (0.0045) | (0.0055) | (0.0057) | (0.0062) |
| FTA | 0.0630 *** | 0.0661 *** | 0.0500 *** | 0.0482 *** | 0.0255 | 0.1237 *** |
| | (0.0052) | (0.0066) | (0.0136) | (0.0145) | (0.0158) | (0.0156) |
| DOP | −0.0276 *** | −0.0250 ** | −0.0192 | −0.1471 *** | −0.0496 ** | 0.0101 |
| | (0.0101) | (0.0118) | (0.0243) | (0.0333) | (0.0246) | (0.0262) |
| LAN | 0.0505 *** | 0.0041 | 0.0060 | 0.1915 *** | 0.0775 *** | 0.0460 * |
| | (0.0091) | (0.0102) | (0.0221) | (0.0429) | (0.0211) | (0.0261) |
| DVA | 0.0118 *** | 0.0128 *** | 0.0006 | 0.0079 | 0.0142 *** | 0.0138 ** |
| | (0.0017) | (0.0021) | (0.0050) | (0.0050) | (0.0040) | (0.0056) |
| EXR | −0.0888 *** | −0.0574 *** | −0.2098 *** | −0.1240 *** | −0.1015 *** | −0.0988 ** |
| | (0.0132) | (0.0170) | (0.0380) | (0.0391) | (0.0366) | (0.0402) |
| DIN | −0.1490 *** | −0.1348 *** | −0.1081 *** | −0.1803 *** | −0.0358 | −0.3041 *** |
| | (0.0125) | (0.0161) | (0.0281) | (0.0394) | (0.0293) | (0.0327) |
| Constant | −3.3570 *** | −3.3481 *** | −3.1667 *** | −3.0244 *** | −3.2665 *** | −4.1200 *** |
| | (0.0443) | (0.0507) | (0.1007) | (0.1575) | (0.0879) | (0.1161) |
| Fixed effect of year | Yes | Yes | Yes | Yes | Yes | Yes |
| Fixed effect of country i | Yes | Yes | Yes | Yes | Yes | Yes |
| Fixed effect of country j | Yes | Yes | Yes | Yes | Yes | Yes |
| Fixed effect of products | Yes | Yes | Yes | Yes | Yes | Yes |
| Observations | 16,397,457 | 9,123,268 | 1,784,998 | 2,050,759 | 1,503,853 | 1,934,579 |
| R-squared | 0.245 | 0.256 | 0.349 | 0.257 | 0.328 | 0.308 |

Note: *** represents significant value at a 1% level, ** represents significant value at a 5% level and * represents significant value at a 1% level. This paper empirically analyzes the standard error clustered to the level of HS6-bit agricultural products.

Regarding the analysis of influence factors of trade complementarity of the participating countries, the results are shown in Table 8. First, the geographical distance coefficient has a significantly negative value at a level of 1%, namely, the geographical distance significantly inhibits the trade complementarity of agricultural products. Distance has a relatively great influence on animal husbandry. Second, the difference coefficient of the development level has a significantly positive value at a level of 1%. Because most of the participating countries are developing countries, and they are at a similar economic development level, these countries effectively promote the trade complementarity of agricultural products. Third, the coefficient of the free trade agreement has a significantly positive value at a level of 1%. On the one hand, signing the agreement may intensify the competition of agricultural products in domestic and foreign markets. On the other hand, it can improve complementarity through the integration of agricultural production services. Similarly, the open-up difference has a significantly negative value at a level of 1%, namely, the larger the open-up differences between two countries, the smaller the possibility of complementarity. Fourth, the common language coefficient has a significantly positive value. We know that the same culture and demand are conducive to technical communication and exchange of agricultural products production to improve trade complementarity. Additionally, the same language environment is also helpful to communication and exchange of agriculture policies of each country, establishment of plans and measures of agriculture cooperation, and resolution of various problems in agriculture cooperation. Fifth, the difference coefficient of proportion of agricultural added value has a significantly positive value at a level of 1%. The larger the difference index, the larger the difference in industrialization processes between the two countries, which also indicates that there is large demand difference between the two countries, namely, different demands on agricultural products are conducive to the promotion of trade complementarity of different agricultural products. Sixth, the exchange rate coefficient has a significantly negative value at a level of 1%. A smaller exchange rate indicates the currency devaluation of country *i*, which is conducive to an increase in the export scale of agricultural products in country *i* and the improvement of trade complementarity. Seventh, the difference coefficient of infrastructure level has a significantly negative value at a level of 1%,indicating that the smaller the differences in in infrastructure, the stronger the complementarity.

### 3.3. Expansion Analysis

This paper focuses on studying the export trade of agricultural products in the participating countries and investigates the influence of the Belt and Road Initiative on each participating country from the perspective of product quality. First of all, China has made a large-scale investment in infrastructure in the participating countries along the Belt and Road Initiative. For economically underdeveloped countries, investments in infrastructure can effectively promote the level of economic development [36]. Through agricultural investments in farmland water conservancy facilities and road infrastructure, China has further improved the agricultural infrastructure of the participating countries, promoted the free flow of agricultural production factors, and reduced the cost of agricultural production factors, which are conducive to R&D and innovation of agricultural production enterprises and the improvement of the quality of agricultural products. Secondly, as a regional free trade agreement, the Belt and Road Initiative can effectively reduce tariff and non-tariff barriers to realize trade liberalization, which further intensifies the competition in domestic and international markets and promotes domestic enterprises to promote the quality of agricultural products to maintain market shares or expand export scales. Such a promotion effect is obviously reflected in economically underdeveloped countries [37]. Finally, under the framework of the initiative, the participating countries actively carry out agricultural technology innovation and cooperation through exchange of such resources as achievements, technology, capital, talents, services, and others to achieve high-quality development of agricultural products [38]. Therefore, this paper further studies the impact of the initiative on the export quality of agricultural products in these participating countries.

### 3.3.1. Quality Calculation of Agricultural Products

This paper mainly learns from the method of Khandelwal et al. [39] to calculate the export quality of agricultural products to move the price and quantity to the left of the equation. The key to this method is assignment to substitution elasticity $\sigma$. Referring to the method of Broada et al. [40], the substitution elasticity $\sigma$ of different HS3-bit code demands of agricultural products imported by different countries is assigned, and the fixed effect of controlling year, destination of exporting country, and product is regressed.

$$\ln q_{ihjt} + \sigma_{Hj}\ln p_{ihjt} = x_h + x_{jt} + \epsilon_{ihjt} \tag{7}$$

where $q$ and $p$, respectively, represent the quantity and price of agricultural product h exported by country $i$ to country $h$ (to be processed digitally). $\sigma_{Hj}$ represents the demand substitution elasticity of HS3-bit agricultural product H of the different destination country $j$. $x_h$ represents the fixation effect of HS6-bit agricultural products, which is used to control the differences between different agricultural products. $x_{jt}$ represents the combined fixed effect of the different destination country $j$ and year $t$, which is used to control the differences caused by the demand preferences of different destination countries. The logarithm of agricultural product quality is obtained by a regression residual of the above equation dividing by $\left(\sigma_{hj} - 1\right)$. In order to compare the qualities of different agricultural products, they are standardized by the maximum $Quality_{ht,max}$ and minimum $Quality_{ht,min}$ of HS6-bit agricultural products exported per year. The amount of the agricultural product quality $h$ exported by country $i$ is standardized by the following formula:

$$Quality_{ih,standard} = \frac{quality_{ih} - quality_{ht,min}}{quality_{ht,max} - quality_{ht,min}} \tag{8}$$

Finally, the quality of the agricultural products exported by the country is weighted on the basis of the amount of the agricultural products exported by the countries to different destination countries to obtain the export quality of agricultural products at a national level.

### 3.3.2. Data Sources and Model Setting

The export data used in this paper are abstracted from the UN COMTRADE database from 2012 to 2020. The codes of agricultural products are uniformly converted into the HS2012 version, and the classification standards of the agricultural products and the screening of the participating countries are consistent with previous sections in this paper. The data of control variables is abstracted from the database of the World Bank.

Because different countries signed the initiative and related documents at different times, this paper investigated the influence of the initiative on the export quality of the agricultural products in the participating countries with reference to Beck et al.'s [41] method. The model is set as follows:

$$Quality_{iht} = \beta_0 + \beta_1 DID_{it} + \sum\nolimits_w \beta_w X_w + \gamma_t + \vartheta_{ih} + \varepsilon_{iht} \tag{9}$$

where $quality_{iht}$ represents the export quality of the agricultural products of the country and $DID_{it}$ represents the virtual variable of the country's participation in the initiative. If the country participates in the initiative in year $t$, the value $DID_{it}$ will be 1 in year $t$ and after, otherwise it will be 0. $X_w$ represents control variables at the national level, mainly including the number of free trade agreements (QFTA) signed by the participating countries, the exchange rate level (EXR) expressed by the indirect pricing method (priced according to one unit of USD), the proportion of value-added agriculture into GDP (PVA), direct foreign investment (FDI), and infrastructure level (synthetic control method: fixed broadband, air cargo traffic, internet penetration rate, wharf water discharge, per capita fixed telephone amount, per capita mobile telephone amount, railway freight volume), and financial development level (synthetic control method: stock market transaction volume, number of listed companies, virtual variables participating in AIIB, degree of national credit

management, gross national savings). The data are consistent with the sources described in Section 3.2. In order to ensure the accuracy of the regression results, the control variables are logarithmized. A descriptive analysis of the regression variables is presented in Table 9. $\gamma_t$ represents the fixed effect at the year level, $\vartheta_{ih}$ represents the combined joint fixed effect between country and the product, and $\varepsilon_{iht}$ represents the random disturbance item.

**Table 9.** Descriptive analysis.

| Variable | Obs | Mean | Std. Dev. | Min | Max |
| --- | --- | --- | --- | --- | --- |
| Quality | 265032 | 0.6015 | 0.2031 | 0 | 1 |
| DID | 265,032 | 0.5312 | 0.4990 | 0 | 1 |
| QFTA | 265045 | 2.7837 | 0.4375 | 1.0986 | 3.3322 |
| EXR | 265045 | 3.0090 | 2.9291 | −1.1929 | 10.2755 |
| PVA | 265045 | 1.9736 | 0.4275 | 1.4794 | 2.8723 |
| FDI | 265045 | 20.6177 | 5.4128 | 0 | 26.2142 |
| INF | 265045 | 1.1452 | 1.1944 | 0.1804 | 8.7936 |

Note: variables above are logarithmic except for dummy variables such as quality and DID.

### 3.3.3. Empirical Results and Analysis

1. Baseline Regression

In this paper, we investigated the impact of the Belt and Road Initiative on the export quality of the agricultural products in the participating countries using a multi-phase double difference method. The method can better solve the endogenous problems caused by taking policy variables as independent variables and can get an unbiased estimator of the policy implementation effect by eliminating unobservable factors that do not change with time [42]. According to the principle of the double difference method, with controlling other influencing factors, the coefficient of explanatory variables $D_{it}$ indicates the change of the quality of the agricultural products after the countries participated in the initiative. Therefore, $DID_{it}$ represents the key recognition coefficient. If the recognition result $> 0$, it is suggested that the country's participation in the initiative significantly promotes the export quality of its agricultural products. Table 10 reports the regression results of multi-phase double difference method.

**Table 10.** Influence of the Belt and Road Initiative on export quality of agricultural products in the participating countries.

| | Export Quality of Agricultural Products | | |
| --- | --- | --- | --- |
| | (1) | (2) | (3) |
| VARIABLES | Quality | Quality | Quality |
| DID | 0.0020 ** | 0.0022 ** | |
| | (0.0010) | (0.0010) | |
| D2012 | | | 0.0004 |
| | | | (0.0014) |
| D2013 | | | 0.0047 ** |
| | | | (0.0020) |
| D2014 | | | 0.0046 * |
| | | | (0.0024) |
| D2015 | | | 0.0143 *** |
| | | | (0.0028) |
| D2016 | | | 0.0094 *** |
| | | | (0.0033) |
| D2017 | | | 0.0181 *** |
| | | | (0.0038) |
| D2018 | | | 0.0143 *** |
| | | | (0.0046) |

**Table 10.** *Cont.*

| | Export Quality of Agricultural Products | | |
| --- | --- | --- | --- |
| | **(1)** | **(2)** | **(3)** |
| D2019 | | | 0.0235 *** |
| | | | (0.0059) |
| D2020 | | | 0.0291 *** |
| | | | (0.0079) |
| QFTA | | 0.0190 *** | 0.0205 *** |
| | | (0.0031) | (0.0032) |
| EXR | | 0.0068 *** | 0.0050 *** |
| | | (0.0017) | (0.0017) |
| PVA | | 0.0079 | 0.0120 * |
| | | (0.0069) | (0.0071) |
| FDI | | −0.0001 ** | −0.0001 ** |
| | | (0.0001) | (0.0001) |
| INF | | 0.0161 *** | 0.0158 *** |
| | | (0.0028) | (0.0029) |
| Fixed effect of time | Yes | Yes | Yes |
| Combined fixed effect of country and product | Yes | Yes | Yes |
| Constant | 0.6043 *** | 0.5110 *** | 0.5023 *** |
| | (0.0005) | (0.0171) | (0.0175) |
| Observations | 254,375 | 254,375 | 254,375 |
| R-squared | 0.717 | 0.718 | 0.718 |

Note: *** represents significant value at a 1% level, ** represents significant value at a 5% level and * represents significant value at a 1% level. This paper empirically analyzes the standard error clustered to the level of HS6-bit agricultural products.

As shown in columns (1) and (2) of Table 10, column (1) only controls the regression results of the combined fixed effects of the year, country, and product, and column (2) controls the control variables at the national level on the basis of column (1). The coefficient of core explanatory variables $DID_{it}$ has a significantly positive value at a level of 5%, which suggests that on the one hand, the initiative has significantly improved the export quality of the agricultural products in the participating countries to achieve the high-quality development. On the other hand, the initiative contributes to the common development of the participating countries from the perspective of the dimension of export quality of the agricultural products.

2.  Parallel Trend Test

Parallel trend means that the individuals in the treatment group, without being treated, have the same time change trend as the individuals in the control group, which is the prerequisite for the correct recognition of causal effects by the double difference method. However, because the counterfactual situation of the individuals of the treatment group after being impacted by the policy (the individuals treated have not been impacted by the policy) failed to be observed, it is generally necessary to test whether the treatment group has the same change trend as the control group before being impacted by the policy so as to indirectly test the parallel trend. Therefore, whether the change trend after the policy impact is the same can be judged by observing whether the treatment group has the same change trend as the control group before the policy impact [43]. Before putting forward the Belt and Road Initiative, there may be other policies to impact the export quality of the agricultural products in the participating countries, so as to result in biased regression results. Therefore, this paper examined the influence of policy changes on regression results except the policy impacts through a parallel trend test, and the results are shown in Figure 2.

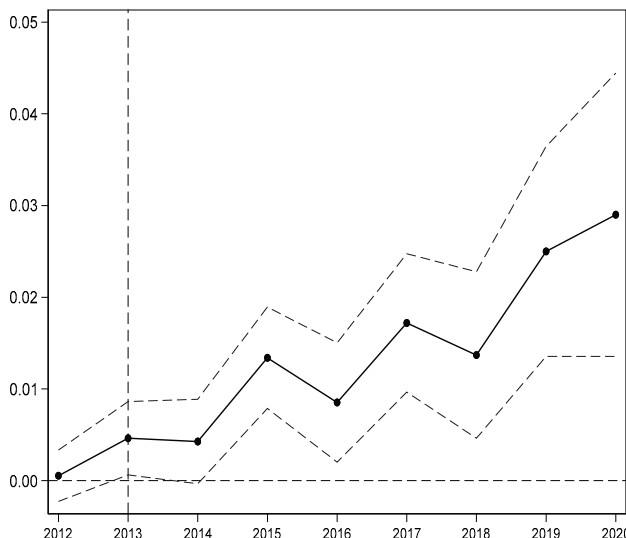

**Figure 2.** Parallel trend test.

In this paper, the dummy variables of one year before the implementation of the policy (because there are a fewer samples of the countries from two years and three years before the implementation of the policy, the observed values of these years are classified as the variables of one year before the implementation of the policy) and eight years after the implementation of the policy (including 2013) are put into the model (9) for regression. The results are shown in column (3) of Table 10. The dummy variable for one year before the implementation of the policy (D2012) is not significant, while the dummy variable for the year of the implementation of the policy has a significantly positive value at a level of 5%. The coefficient gradually increases after that, indicating that the initiative has a long-term improvement effect on the export quality of agricultural products in these participating countries. For presentation of the results, this paper draws a parallel trend test chart based on the results in Table 10. The dotted line in the chart represents the confidence interval (±5%), the solid line is the line chart of dummy variable coefficients of different years, and the zero value only passes through the confidence interval of regression coefficients in 2012. The model passes the parallel trend test.

3. Placebo Test

The placebo test aims to test whether the regression results are affected by other political or random factors by constructing re-regression of virtual treatment groups or virtual policy impact time. In order to further test the influence of omitted variables such as other political or random factors on the regression results of the multi-phase double difference method. A replacement test is employed in this paper to draw the placebo test chart as shown in Figure 3.

Because of the multi-phase double difference method, a double placebo test with time and individual randomization is employed in this paper. Time randomization, in this paper, means that one year was randomly selected as the year when the policy is implemented to investigate whether the benchmark results of this paper are caused by other non-synchronous factors. Individual randomization, meaning that one individual was randomly selected as the object to which the policy was implemented to investigate whether the benchmark results of this paper were caused by other policy factors in the same period. This test method was done by randomly selecting certain countries as test group in each year with random impact [44]. Regardless of 2012 and 2020, we randomly selected four years from 2013 to 2019 as the time group when the policy was implemented, which aimed to ensure that there are samples for comparison before and after the test. In the year $t_1$, $n_1$ countries were randomly selected without replacement as the countries participating in the Belt and Road Initiative. In the year $t_2$, $n_2$ countries were randomly selected without replacement as the countries participating in the initiative; in the year $t_3$, $n_3$ countries were

randomly selected without replacement as the countries participating in the initiative; in the year $t_4$, $n_4$ countries were randomly selected without replacement as the countries participating in the initiative, to ensure that the sum of $n_1, n_2, n_3, and\ n_4$ was equal to the countries actually participating in the initiative. Figure 3 shows the regression results of 500 random samples. The solid line in the figure is the standard normal distribution, the dotted line is the kernel density map of the regression coefficients of the 500 samples, and the small circle below is the *p* value of the core explanatory variables of regression of the 500 samples, so that we could find that the regression coefficients and *p* value were distributed near 0 and close to the normal distribution. At the same time, the benchmark regression coefficient of 0.0022 was relatively far from the center value 0, which suggested that the influence of the Belt and Road Initiative on the export quality of agricultural products has not been disturbed by missing variables, and a robust regression result is obtained in this paper.

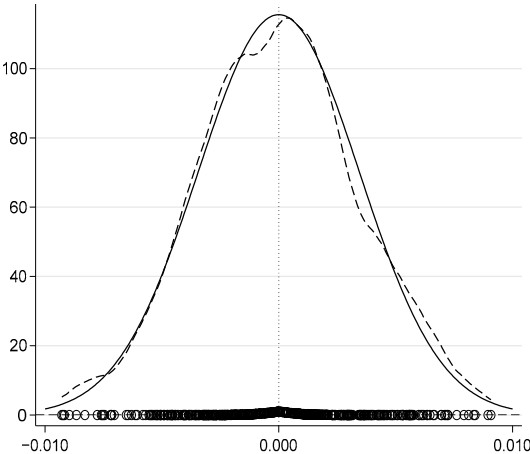

**Figure 3.** Placebo test.

## 4. Cluster Analysis and Heterogeneity Analysis

In this study, the RCA, ESI, and TCI are utilized as indicators for cluster analysis, and hierarchical cluster processing was performed on a sample of 65 countries. The NbClust package was employed to determine the optimal number of clusters (see Figure 4), and the hclust function was utilized to conduct hierarchical clustering of the samples, which was then visualized in the form of a cluster diagram (see Figure 5). The results of the hierarchical clustering indicate that the optimal number of clusters for the sample of 65 countries is seven, which corresponds to the geographical divisions. It is observed that China's RCA, ESI, and TCI differ somewhat from those of other countries, and therefore, China is excluded from the heterogeneity analysis. The final result created six distinct groups.

Using the cluster analysis results, this paper conducted a heterogeneity analysis to investigate the impact of the Belt and Road Initiative on the quality of agricultural products exported by countries along the route.

The results of the heterogeneity analysis regression, as presented in Table 11, indicate that the Belt and Road Initiative has a positive impact on promoting the quality improvement of agricultural products in various regional countries. The promotion effect is particularly significant for agricultural products exported by countries in Central Asia and South Asia due to their large agricultural scale, which makes the dissemination and exchange of agricultural technology more effective under the Belt and Road framework. However, the effect of promoting the quality of agricultural exports in other regions is not significant.

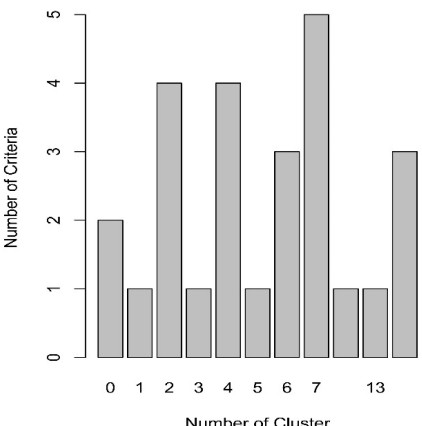

**Figure 4.** Clustered voting map.

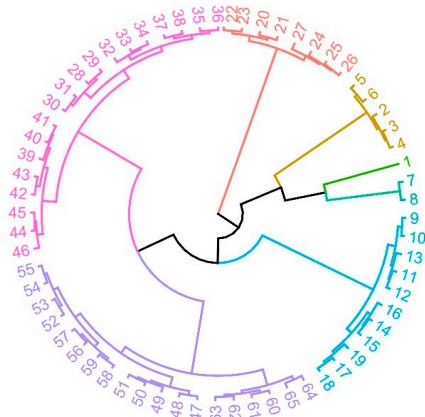

**Figure 5.** Cluster analysis chart. Notes: Country numbering is consistent with Table A4.

**Table 11.** Heterogeneity analysis of product quality mentions for agricultural exports.

| | Central Asia (1) | Northeast Asia (2) | Southeast Asia (3) | South Asia (4) | Central and Eastern Europe (5) | South Asia Middle East (6) |
|---|---|---|---|---|---|---|
| VARIABLES | Quality | Quality | Quality | Quality | Quality | Quality |
| DID | 0.0137 ** | 0.0090 | 0.0020 | 0.0090 *** | 0.0012 | 0.0020 |
| | (0.0067) | (0.0336) | (0.0028) | (0.0029) | (0.0014) | (0.0012) |
| QFTA | 0.0077 | −0.0688 | 0.0226 *** | 0.0890 *** | 0.0174 *** | −0.0059 ** |
| | (0.0275) | (0.1084) | (0.0061) | (0.0236) | (0.0042) | (0.0029) |
| EXR | 0.0288 *** | −0.0434 | −0.0035 | 0.0174 | −0.0004 | −0.0012 |
| | (0.0108) | (0.0787) | (0.0081) | (0.0168) | (0.0017) | (0.0016) |
| PVA | 0.0128 | 0.1109 ** | 0.0212 *** | 0.0011 | 0.0018 | 0.0079 *** |
| | (0.0100) | (0.0476) | (0.0052) | (0.0058) | (0.0033) | (0.0025) |
| FDI | 0.0009 ** | 0.0000 | 0.0002 | 0.0007 | 0.0000 | −0.0001 |
| | (0.0004) | (0.0007) | (0.0002) | (0.0010) | (0.0001) | (0.0001) |
| INF | 0.0956 *** | 0.0092 | 0.0172 *** | 0.0269 ** | 0.0087 ** | 0.0259 *** |
| | (0.0175) | (0.0765) | (0.0040) | (0.0116) | (0.0043) | (0.0032) |
| Fixed effect of time | Yes | Yes | Yes | Yes | Yes | Yes |
| Combined fixed effect of country and product | Yes | Yes | Yes | Yes | Yes | Yes |
| Constant | 0.3760 *** | 0.4555 *** | 0.7341 * | 0.7173 *** | 0.5228 *** | 0.5414 *** |
| | (0.0990) | (0.0538) | (0.4307) | (0.1023) | (0.0129) | (0.0092) |

**Table 11.** *Cont.*

| | Central Asia (1) | Northeast Asia (2) | Southeast Asia (3) | South Asia (4) | Central and Eastern Europe (5) | South Asia Middle East (6) |
|---|---|---|---|---|---|---|
| Observations | 10,367 | 7552 | 44,573 | 24,629 | 92,000 | 68,884 |
| R-squared | 0.667 | 0.722 | 0.644 | 0.724 | 0.639 | 0.572 |

Note: *** represents significant value at a 1% level, ** represents significant value at a 5% level and * represents significant value at a 1% level. This paper empirically analyzes the standard error clustered to the level of HS6-bit agricultural products.

## 4. Discussion

In the initial phase of cooperation, countries along the Belt and Road exhibit greater similarity in their agricultural exports. However, as policy exchange deepens, the competitiveness of agricultural exports of the participating countries along the Belt and Road shows a trend of gradual dispersion and decline, ultimately demonstrating strong export complementarity. This phenomenon has been corroborated by Liu et al. who, through social network analysis, have demonstrated that competition and complementarity coexist in the agriculture of countries along the Belt and Road. Additionally, they have established that complementarity in trade outweighs export competition [11].

The adherence of China and the countries along the route to the principle of five links, which includes political communication, linkage of facilities, smooth trade, financial integration, and people-to-people contacts, has contributed to the improvement of trade facilitation and the quality of agricultural exports in the region. Fan et al. have provided evidence supporting this claim by demonstrating the positive impact of trade facilitation on agricultural exports using China and ASEAN countries as examples [45].

The geographic distance between countries along the trade route impedes competitiveness and complementarity to a significant extent, whereas reducing cultural distance can effectively enhance complementarity while decreasing competitiveness. This finding has been corroborated by Xing et al. who conducted a study on the impact of multidimensional distances on China's agricultural exports, using various measures such as geographical, cultural, economic, and institutional distances [46]. Their analysis revealed that geographical distance represents a major obstacle to agricultural trade, while decreasing cultural distance has the potential to promote the export of high-value agricultural products.

FTAs have had a significant impact on the competitiveness and complementarity of agricultural products, contrary to the findings of Narayan and Bhattacharya's research. Their study investigated the determinants that affect the relative export competitiveness (REC) of Indian agricultural products and concluded that the World Trade Organization (WTO) has a positive impact on the REC of rice, while the South Asian Free Trade Area (SAFTA) agreement has a negative impact on the REC of wheat and rice. This is mainly due to the vertically linked and concentrated nature of developed country food markets and the presence of non-tariff measures in agriculture in various countries [33].

This study concludes that agricultural exports' competitiveness is reduced by an appreciation of the exchange rate, while a depreciation of the exchange rate increases the complementarity of agricultural exports. These findings differ from those of Kandilov, who studied the effect of exchange rate volatility on agricultural exports using a sample of agricultural exports from 69 countries classified by the IMF as developed, emerging, and developing. His research discovered that exchange rate volatility has a considerable negative impact on agricultural trade between nations and that this effect is more pronounced for developing country exporters [47].

While the sample data in this study extends to 2020, it is important to note that the current economic landscape is marked by a multitude of uncertain events. Therefore, future research ought to expand the sample range further and conduct an in-depth analysis of the impact of economic uncertainty on agricultural trade.

In light of the open platform offered by the Belt and Road Initiative, it is crucial to establish uniform quality and safety standards for agricultural products. This will enable the creation of a transparent, open, and fair agricultural trade environment. Moreover, the major agricultural countries along the route should bolster their agricultural trade cooperation, reduce the cost of trade in agricultural products, promote organic farming and modern agricultural construction, and thus contribute to the high-quality development of agricultural products.

## 5. Conclusions

This paper employs the UN COMTRADE database from 2012 to 2020 to calculate the revealed comparative advantage index, export similarity index, and trade complementarity index of the countries participating along the Belt and Road. Subsequently, the gravity model is used to investigate the factors influencing the export similarity of agricultural products and trade complementarity. Moreover, the multi-period differences-in-differences method is used to explore the impact of the Belt and Road Initiative on the quality of agricultural product exports in the participating countries. The findings provide substantial support for the high-quality development of agricultural products in these countries.

The main findings of this study include: First, the complementarity of agricultural trade between participating countries along the Belt and Road is high, and there is more room for trade cooperation. Second, distance is an important factor hindering the competitiveness and complementarity of agricultural exports. Free trade agreements are an important factor in improving the competitiveness and complementarity of agricultural products. Countries that are more open to the outside world are relatively more competitive in agricultural products. Complementarity in agricultural trade is relatively greater in countries with large differences in the degree of openness. A similar cultural background reduces the competitiveness of agricultural exports and increases complementarity in agricultural trade. Countries with small differences in agricultural value added have relatively greater competitiveness in agricultural products. Countries with large differences in agricultural value added have relatively greater complementarity in agricultural products. Currency appreciation reduces the competitiveness of agricultural exports and currency depreciation increases the complementarity of agricultural products. Lower infrastructure differentials increase the competitiveness of agricultural exports and trade complementarity. Third, the Belt and Road Initiative has significantly improved the export quality of agricultural products from countries along the route, leading to high-quality development of agricultural products.

This paper proposes the following countermeasures: First, participating countries along the Belt and Road should participate in international trade according to their comparative advantages in agricultural products in order to improve the international competitiveness of agricultural products. At the same time, they should import agricultural products according to their comparative disadvantages in order to increase the complementarity of agricultural products. Second, countries along the Belt and Road should attach great importance to signing and implementing bilateral policies to achieve political communication and people-to-people exchanges. Countries along the Belt and Road should maintain some of the trade barriers for agricultural products to avoid excessive trade liberalization that leads to blind competition. Third, participating countries along the Belt and Road should stabilize the value of their currencies to avoid the negative impact of currency fluctuations on agricultural exports. Countries should also strengthen infrastructural connectivity to reduce logistics and transport costs as well as information asymmetries in the production of agricultural products. Fourth, depending on the endowments of participating countries along the Belt and Road, we should promote the cross-border flow of production factors. We will attract the participation of developed countries with relatively large amounts of capital and technology to achieve high-quality development in the participating countries along the Belt and Road.

In the context of economic globalization, trade relations between countries have gradually relied on the division of the value chain. Therefore, analysis of trade relations between nations based on competitiveness, complementarity, and similarity of trade warrants an expanded scope. As agriculture in countries along the route continues to evolve, the value chain for agricultural products becomes increasingly sophisticated. Future research may benefit from considering the value chain of agricultural products as a key perspective.

**Author Contributions:** Conceptualization, X.W.; methodology, J.L. (Jia Li); software, Y.C. and J.S.; validation, J.L. (Jianxu Liu); formal analysis, J.S.; investigation, and Y.C.; resources, S.S.; data curation, J.S.; writing—original draft preparation, X.W.; writing—review and editing, J.S.; visualization, J.L. (Jia Li); supervision, J.L. (Jianxu Liu); funding acquisition, S.S. All authors have read and agreed to the published version of the manuscript.

**Funding:** This research received no external funding.

**Institutional Review Board Statement:** Not applicable.

**Informed Consent Statement:** Not applicable.

**Data Availability Statement:** All data can be obtained by email from the corresponding author.

**Acknowledgments:** This work has been assisted by financial support from the China-ASEAN High-Quality Development Research Center at Shandong University of Finance and Economics and the "Theoretical Economics Research Innovation Team" of the Youth Innovation Talent Introduction and Education Plan of Colleges and Universities in Shandong Province.

**Conflicts of Interest:** The authors declare no conflict of interest.

## Appendix A

**Table A1.** RCA Index of agricultural products in the participating countries (or regions) along the Belt and Road Initiative (schedule).

| Type | Year | China | Central Asia | Northeast Asia | Southeast Asia | South Asia | Central and Eastern Europe | South Asia, Middle East |
|---|---|---|---|---|---|---|---|---|
| Planting industry | 2012 | 0.7771 | 1.8389 | 0.9369 | 0.9620 | 1.4327 | 1.0178 | 1.4210 |
| | 2013 | 0.8074 | 1.8358 | 3.4084 | 0.9450 | 1.3638 | 1.0050 | 1.4050 |
| | 2014 | 0.7821 | 1.7323 | 0.1342 | 0.9653 | 1.3717 | 1.0212 | 1.4007 |
| | 2015 | 0.7838 | 1.7729 | 0.5124 | 0.9663 | 1.3596 | 1.0302 | 1.4010 |
| | 2016 | 0.8056 | 1.7733 | 1.4100 | 0.9704 | 1.3234 | 1.0251 | 1.4045 |
| | 2017 | 0.8141 | 1.7541 | 0.7945 | 1.0147 | 1.2864 | 0.9940 | 1.3907 |
| | 2018 | 0.8103 | 1.7989 | 3.7217 | 0.9919 | 1.3284 | 0.9815 | 1.4281 |
| | 2019 | 0.8452 | 1.7970 | 0.1599 | 0.9793 | 1.3047 | 1.0193 | 1.4724 |
| | 2020 | 0.8654 | 1.3903 | 0.6009 | 0.9856 | 1.4219 | 0.9896 | 1.2885 |
| Forest industry | 2012 | 1.5164 | 0.0904 | 1.6462 | 1.8218 | 0.1228 | 1.7005 | 0.3930 |
| | 2013 | 1.4832 | 0.0501 | 0.9490 | 1.8837 | 0.1030 | 1.7215 | 0.3535 |
| | 2014 | 1.5071 | 0.0246 | 3.2195 | 1.8389 | 0.1171 | 1.6920 | 0.3294 |
| | 2015 | 1.5803 | 0.0169 | 0.1298 | 1.7010 | 0.1477 | 1.6818 | 0.2967 |
| | 2016 | 1.5074 | 0.0768 | 0.5529 | 1.6131 | 0.1484 | 1.6983 | 0.2823 |
| | 2017 | 1.4578 | 0.0978 | 1.3958 | 1.6165 | 0.1554 | 1.7823 | 0.3221 |
| | 2018 | 1.4560 | 0.1369 | 0.8998 | 1.5046 | 0.1578 | 1.7864 | 0.3126 |
| | 2019 | 1.4285 | 0.1251 | 3.3172 | 1.5151 | 0.2009 | 1.7281 | 0.2905 |
| | 2020 | 1.4415 | 0.2275 | 0.1397 | 0.9625 | 0.4305 | 1.4144 | 0.5760 |
| Animal husbandry | 2012 | 0.6577 | 0.0994 | 0.5354 | 0.1872 | 0.4994 | 1.1678 | 0.7601 |
| | 2013 | 0.5866 | 0.1405 | 1.5305 | 0.1912 | 0.5688 | 1.1647 | 0.8448 |
| | 2014 | 0.6287 | 0.4782 | 0.9549 | 0.2004 | 0.5812 | 1.1011 | 0.8552 |
| | 2015 | 0.6204 | 0.2498 | 3.1188 | 0.2174 | 0.6258 | 1.1096 | 0.8696 |
| | 2016 | 0.5828 | 0.2385 | 0.1507 | 0.2180 | 0.6407 | 1.1309 | 0.8339 |
| | 2017 | 0.6004 | 0.2984 | 0.4970 | 0.2042 | 0.5711 | 1.1769 | 0.8336 |
| | 2018 | 0.6210 | 0.3485 | 1.4252 | 0.2371 | 0.5342 | 1.1762 | 0.8182 |

**Table A1.** *Cont.*

| Type | Year | China | Central Asia | Northeast Asia | Southeast Asia | South Asia | Central and Eastern Europe | South Asia, Middle East |
|------|------|-------|--------------|----------------|----------------|------------|-----------------------------|-------------------------|
| | 2019 | 0.6814 | 0.4233 | 0.9485 | 0.2380 | 0.5492 | 1.0970 | 0.6805 |
| | 2020 | 0.4919 | 0.6126 | 3.3771 | 0.5333 | 0.4671 | 1.0241 | 0.9942 |
| Agricultural and sideline industry | 2012 | 0.8648 | 0.2442 | 0.1726 | 0.7056 | 0.5891 | 0.7312 | 0.6222 |
| | 2013 | 0.7952 | 0.2724 | 0.4820 | 0.7825 | 0.6525 | 0.7369 | 0.6282 |
| | 2014 | 0.8738 | 0.2979 | 1.3492 | 0.7910 | 0.5127 | 0.7394 | 0.6616 |
| | 2015 | 0.8923 | 0.2869 | 1.0508 | 0.8142 | 0.4156 | 0.7294 | 0.6119 |
| | 2016 | 0.9409 | 0.2788 | 2.8529 | 0.8443 | 0.3778 | 0.7322 | 0.6212 |
| | 2017 | 0.9218 | 0.2956 | 0.1635 | 0.8130 | 0.4702 | 0.7483 | 0.6433 |
| | 2018 | 0.8989 | 0.2500 | 0.3951 | 0.9244 | 0.4878 | 0.7707 | 0.5909 |
| | 2019 | 0.8540 | 0.2397 | 1.3710 | 0.9787 | 0.4677 | 0.7810 | 0.5872 |
| | 2020 | 0.7987 | 0.6410 | 0.9169 | 1.2085 | 0.5039 | 0.9799 | 0.7127 |
| Aquaculture | 2012 | 1.6447 | 0.2648 | 3.1705 | 2.2629 | 1.0700 | 0.4787 | 0.4215 |
| | 2013 | 1.6819 | 0.3264 | 0.2167 | 2.1865 | 1.2916 | 0.5045 | 0.4443 |
| | 2014 | 1.6454 | 0.3346 | 0.4851 | 2.0665 | 1.4857 | 0.5290 | 0.4535 |
| | 2015 | 1.5894 | 0.3472 | 1.6171 | 1.9603 | 1.5216 | 0.5212 | 0.4671 |
| | 2016 | 1.5540 | 0.2951 | 1.0308 | 1.8938 | 1.6836 | 0.5200 | 0.5024 |
| | 2017 | 1.5772 | 0.2877 | 1.7146 | 1.8980 | 1.7880 | 0.5272 | 0.4981 |
| | 2018 | 1.5972 | 0.1912 | 0.6304 | 1.8304 | 1.6956 | 0.5319 | 0.5278 |
| | 2019 | 1.5038 | 0.1650 | 0.7934 | 1.8494 | 1.8017 | 0.5270 | 0.5667 |
| | 2020 | 1.6426 | 0.2804 | 1.1845 | 1.4771 | 1.1114 | 0.6921 | 0.4434 |

**Table A2.** Export similarity of agricultural products in the participating countries (or regions) along the Belt and Road Initiative (schedule).

| Year | State | State | Planting Industry | Forest Industry | Animal Husbandry | Agricultural and Sideline Industry | Aquaculture |
|------|-------|-------|-------------------|-----------------|------------------|-------------------------------------|-------------|
| 2013 | China | 5 Central Asian countries | 0.4142 | 0.0083 | 0.0210 | 0.0409 | 0.0588 |
| | | Mongolia | 0.3907 | 0.1092 | 0.0402 | 0.0903 | 0.0842 |
| | | 11 Southeast Asian countries | 0.3960 | 0.1328 | 0.0266 | 0.0596 | 0.1234 |
| | | 8 South Asian countries | 0.3243 | 0.0360 | 0.0283 | 0.0561 | 0.0985 |
| | | 19 Central and Eastern European countries | 0.3649 | 0.1369 | 0.0544 | 0.0849 | 0.0539 |
| | | Southwest Asia, 19 Middle Eastern countries | 0.3884 | 0.0281 | 0.0509 | 0.0723 | 0.0508 |
| 2016 | China | 5 Central Asian countries | 0.4258 | 0.0093 | 0.0318 | 0.0313 | 0.0556 |
| | | Mongolia | 0.4258 | 0.1071 | 0.0355 | 0.0629 | 0.0815 |
| | | 11 Southeast Asian countries | 0.3956 | 0.1357 | 0.0272 | 0.0729 | 0.1246 |
| | | 8 South Asian countries | 0.3569 | 0.0358 | 0.0216 | 0.0604 | 0.1035 |
| | | 19 Central and Eastern European countries | 0.3785 | 0.1373 | 0.0481 | 0.0918 | 0.0557 |
| | | Southwest Asia, 19 Middle Eastern countries | 0.4019 | 0.0254 | 0.0440 | 0.0837 | 0.0510 |

**Table A2.** *Cont.*

| Year | State | State | Planting Industry | Forest Industry | Animal Husbandry | Agricultural and Sideline Industry | Aquaculture |
|------|-------|-------|-------------------|-----------------|------------------|-----------------------------------|-------------|
| 2019 | China | 5 Central Asian countries | 0.4348 | 0.0118 | 0.0345 | 0.0358 | 0.0162 |
| | | Mongolia, Russia | 0.2810 | 0.0977 | 0.0463 | 0.0611 | 0.1899 |
| | | 11 Southeast Asian countries | 0.4011 | 0.1209 | 0.0352 | 0.0735 | 0.1177 |
| | | 8 South Asian countries | 0.3696 | 0.0281 | 0.0236 | 0.0713 | 0.1124 |
| | | 19 Central and Eastern European countries | 0.3693 | 0.1303 | 0.0563 | 0.0959 | 0.0600 |
| | | Southwest Asia, 19 Middle Eastern countries | 0.4021 | 0.0168 | 0.0487 | 0.0742 | 0.0699 |
| 2013 | Mongolia, Russia | 5 Central Asian countries | 0.3907 | 0.0122 | 0.0256 | 0.0425 | 0.0335 |
| | | 11 Southeast Asian countries | 0.5067 | 0.0124 | 0.0144 | 0.0335 | 0.0404 |
| | | 8 South Asian countries | 0.5039 | 0.0103 | 0.0160 | 0.0312 | 0.0358 |
| | | 19 Central and Eastern European countries | 0.4808 | 0.0131 | 0.0295 | 0.0446 | 0.0305 |
| | | Southwest Asia, 19 Middle Eastern countries | 0.5734 | 0.0154 | 0.0286 | 0.0391 | 0.0285 |
| 2016 | Mongolia, Russia | 5 Central Asian countries | 0.5620 | 0.0157 | 0.0403 | 0.0288 | 0.0272 |
| | | 11 Southeast Asian countries | 0.5358 | 0.0131 | 0.0230 | 0.0268 | 0.0391 |
| | | 8 South Asian countries | 0.5272 | 0.0110 | 0.0198 | 0.0246 | 0.0360 |
| | | 19 Central and Eastern European countries | 0.4951 | 0.0158 | 0.0455 | 0.0307 | 0.0231 |
| | | Southwest Asia, 19 Middle Eastern countries | 0.5987 | 0.0170 | 0.0425 | 0.0288 | 0.0221 |
| 2019 | Mongolia, Russia | 5 Central Asian countries | 0.2850 | 0.0197 | 0.0468 | 0.0326 | 0.0162 |
| | | 11 Southeast Asian countries | 0.5362 | 0.0123 | 0.0239 | 0.0301 | 0.0142 |
| | | 8 South Asian countries | 0.5329 | 0.0106 | 0.0186 | 0.0298 | 0.0135 |
| | | 19 Central and Eastern European countries | 0.4639 | 0.0197 | 0.0480 | 0.0354 | 0.0153 |
| | | Southwest Asia, 19 Middle Eastern countries | 0.6168 | 0.0165 | 0.0412 | 0.0304 | 0.0142 |
| 2013 | 5 Central Asian countries | 11 Southeast Asian countries | 0.3781 | 0.0870 | 0.0228 | 0.0634 | 0.0529 |
| | | 8 South Asian countries | 0.3067 | 0.0295 | 0.0274 | 0.0592 | 0.0447 |
| | | 19 Central and Eastern European countries | 0.3483 | 0.0870 | 0.0814 | 0.0914 | 0.0302 |
| | | Southwest Asia, 19 Middle Eastern countries | 0.3710 | 0.0182 | 0.0963 | 0.0767 | 0.0277 |

**Table A2.** *Cont.*

| Year | State | State | Planting Industry | Forest Industry | Animal Husbandry | Agricultural and Sideline Industry | Aquaculture |
|------|-------|-------|-------------------|-----------------|------------------|-----------------------------------|-------------|
| 2016 | 5 Central Asian countries | 11 Southeast Asian countries | 0.4665 | 0.0792 | 0.0274 | 0.0492 | 0.0565 |
| | | 8 South Asian countries | 0.4340 | 0.0226 | 0.0236 | 0.0436 | 0.0485 |
| | | 19 Central and Eastern European countries | 0.4395 | 0.0827 | 0.0688 | 0.0593 | 0.0340 |
| | | Southwest Asia, 19 Middle Eastern countries | 0.4892 | 0.0174 | 0.0734 | 0.0545 | 0.0311 |
| 2019 | 5 Central Asian countries | 11 Southeast Asian countries | 0.2667 | 0.0699 | 0.0315 | 0.0468 | 0.1175 |
| | | 8 South Asian countries | 0.2440 | 0.0186 | 0.0230 | 0.0466 | 0.1138 |
| | | 19 Central and Eastern European countries | 0.2502 | 0.0835 | 0.0826 | 0.0584 | 0.0607 |
| | | Southwest Asia, 19 Middle Eastern countries | 0.2675 | 0.0129 | 0.0785 | 0.0487 | 0.0703 |
| 2013 | 11 Southeast Asian countries | 8 south Asian countries | 0.3743 | 0.0332 | 0.0171 | 0.0506 | 0.0619 |
| | | 19 Central and Eastern European countries | 0.3992 | 0.1045 | 0.0276 | 0.0709 | 0.0373 |
| | | Southwest Asia, 19 Middle Eastern countries | 0.4410 | 0.0245 | 0.0264 | 0.0617 | 0.0356 |
| 2016 | 11 Southeast Asian countries | 8 South Asian countries | 0.4073 | 0.0269 | 0.0165 | 0.0590 | 0.0692 |
| | | 19 Central and Eastern European countries | 0.4097 | 0.1015 | 0.0322 | 0.0789 | 0.0417 |
| | | Southwest Asia, 19 Middle Eastern countries | 0.4596 | 0.0224 | 0.0299 | 0.0748 | 0.0382 |
| 2019 | 11 Southeast Asian countries | 8 South Asian countries | 0.4133 | 0.0237 | 0.0175 | 0.0653 | 0.0729 |
| | | 19 Central and Eastern European countries | 0.3890 | 0.0938 | 0.0355 | 0.0832 | 0.0423 |
| | | Southwest Asia, 19 Middle Eastern countries | 0.4573 | 0.0155 | 0.0313 | 0.0637 | 0.0473 |
| 2013 | 8 South Asian countries | 19 Central and Eastern European countries | 0.3534 | 0.0322 | 0.0344 | 0.0701 | 0.0340 |
| | | Southwest Asia, 19 Middle Eastern countries | 0.4072 | 0.0122 | 0.0327 | 0.0628 | 0.0352 |
| 2016 | 8 South Asian countries | 19 Central and Eastern European countries | 0.3822 | 0.0273 | 0.0289 | 0.0644 | 0.0370 |
| | | Southwest Asia, 19 Middle Eastern countries | 0.4384 | 0.0124 | 0.0261 | 0.0652 | 0.0380 |

**Table A2.** *Cont.*

| Year | State | State | Planting Industry | Forest Industry | Animal Husbandry | Agricultural and Sideline Industry | Aquaculture |
|---|---|---|---|---|---|---|---|
| 2019 | 8 South Asian countries | 19 Central and Eastern European countries | 0.3675 | 0.0246 | 0.0268 | 0.0793 | 0.0417 |
| | | Southwest Asia, 19 Middle Eastern countries | 0.4418 | 0.0112 | 0.0232 | 0.0633 | 0.0489 |
| 2013 | 19 Central and Eastern European countries | Southwest Asia, 19 Middle Eastern countries | 0.4154 | 0.0257 | 0.0882 | 0.0872 | 0.0280 |
| 2016 | 19 Central and Eastern European countries | Southwest Asia, 19 Middle Eastern countries | 0.4321 | 0.0231 | 0.0731 | 0.0872 | 0.0283 |

**Table A3.** Trade complementarity index of agricultural products in the participating countries (or regions) along the Belt and Road Initiative (schedule).

| Year | State | State | Planting Industry | Forest Industry | Animal Husbandry | Agricultural and Sideline Industry | Aquaculture |
|---|---|---|---|---|---|---|---|
| 2012 | China | 5 Central Asian countries | 0.1976 | 0.4773 | 0.5833 | 0.5785 | 0.5657 |
| | | Mongolia, Russia | 0.3000 | 0.5482 | 0.5977 | 0.9998 | 0.9086 |
| | | 11 Southeast Asian countries | 0.5204 | 0.9928 | 0.8755 | 0.6722 | 0.9999 |
| | | 8 South Asian countries | 0.5011 | 0.8954 | 0.9996 | 0.5069 | 0.9978 |
| | | 19 Central and Eastern European countries | 0.3660 | 0.7600 | 0.6450 | 0.6291 | 0.9887 |
| | | Southwest Asia, 19 Middle Eastern countries | 0.5034 | 0.5370 | 0.8744 | 0.6460 | 0.6188 |
| 2020 | China | 5 Central Asian countries | 0.4344 | 0.5069 | 0.5281 | 0.8479 | 0.5157 |
| | | Mongolia, Russia | 0.9560 | 0.6350 | 0.5914 | 0.7810 | 0.7575 |
| | | 11 Southeast Asian countries | 0.5905 | 0.7846 | 0.5303 | 0.7584 | 0.7765 |
| | | 8 South Asian countries | 0.3591 | 0.9967 | 0.4960 | 0.7529 | 0.5064 |
| | | 19 Central and Eastern European countries | 0.5306 | 0.6208 | 0.5595 | 0.8638 | 0.7525 |
| | | Southwest Asia, 19 Middle Eastern countries | 0.5788 | 0.5191 | 0.7019 | 0.5011 | 0.5124 |
| 2012 | 5 Central Asian countries | China | 0.4769 | 0.5289 | 0.5479 | 0.8134 | 0.5236 |
| | | Mongolia, Russia | 0.5580 | 0.8717 | 0.7350 | 0.7773 | 0.5915 |
| | | 11 Southeast Asian countries | 0.4575 | 0.9998 | 1.0000 | 0.9397 | 0.4879 |
| | | 8 South Asian countries | 0.5073 | 1.0000 | 0.5713 | 0.5914 | 0.5028 |

**Table A3.** *Cont.*

| Year | State | State | Planting Industry | Forest Industry | Animal Husbandry | Agricultural and Sideline Industry | Aquaculture |
|------|-------|-------|-------------------|-----------------|------------------|-----------------------------------|-------------|
| | | 19 Central and Eastern European countries | 0.6349 | 0.8364 | 1.0000 | 0.8979 | 0.9964 |
| | | Southwest Asia, 19 Middle Eastern countries | 0.9877 | 1.0000 | 1.0000 | 0.6592 | 0.5705 |
| 2020 | 5 Central Asian countries | China | 0.2522 | 0.8496 | 0.5272 | 0.6667 | 0.5243 |
| | | Mongolia, Russia | 0.6941 | 0.5094 | 0.5453 | 0.7283 | 0.5706 |
| | | 11 Southeast Asian countries | 0.5304 | 0.5000 | 0.6850 | 0.6860 | 0.5542 |
| | | 8 South Asian countries | 0.5021 | 1.0000 | 0.8459 | 0.9936 | 0.5491 |
| | | 19 Central and Eastern European countries | 0.3700 | 0.8627 | 0.5415 | 0.9997 | 0.5378 |
| | | Southwest Asia, 19 Middle Eastern countries | 0.8115 | 0.6455 | 0.5059 | 0.7854 | 0.5748 |
| 2012 | Mongolia, Russia | China | 0.5804 | 0.7423 | 0.5012 | 0.6938 | 0.6502 |
| | | 5 Central Asian countries | 0.2694 | 0.5945 | 0.5117 | 0.8555 | 0.6101 |
| | | 11 Southeast Asian countries | 1.0000 | 0.9993 | 0.8639 | 0.5069 | 0.6109 |
| | | 8 South Asian countries | 0.6261 | 0.9998 | 1.0000 | 0.8097 | 1.0000 |
| | | 19 Central and Eastern European countries | 0.4546 | 0.7262 | 0.7493 | 0.9708 | 0.9846 |
| | | Southwest Asia, 19 Middle Eastern countries | 0.8526 | 0.9995 | 0.9749 | 0.7532 | 0.5473 |
| 2020 | Mongolia, Russia | China | 0.3539 | 0.7608 | 0.7467 | 0.8220 | 0.6069 |
| | | 5 Central Asian countries | 0.9992 | 1.0000 | 0.8522 | 0.8821 | 0.5329 |
| | | 11 Southeast Asian countries | 0.5004 | 0.5232 | 0.7036 | 0.8802 | 0.5008 |
| | | 8 South Asian countries | 0.4996 | 0.4976 | 0.5171 | 0.8100 | 0.8798 |
| | | 19 Central and Eastern European countries | 0.4497 | 0.8253 | 1.0000 | 0.8811 | 0.6151 |
| | | Southwest Asia, 19 Middle Eastern countries | 0.9385 | 0.5069 | 0.5140 | 0.8127 | 0.5554 |
| 2012 | 11 Southeast Asian countries | China | 0.4982 | 0.6504 | 0.5551 | 0.7082 | 0.6674 |
| | | 5 Central Asian countries | 0.5486 | 0.6053 | 0.9970 | 0.9991 | 0.5836 |
| | | Mongolia, Russia | 0.5210 | 0.9618 | 0.5026 | 0.8967 | 0.4999 |
| | | 8 South Asian countries | 0.9318 | 0.8962 | 0.5055 | 0.8901 | 0.7387 |
| | | 19 Central and Eastern European countries | 0.6829 | 0.7185 | 0.6892 | 0.6619 | 0.5567 |
| | | Southwest Asia, 19 Middle Eastern countries | 0.6674 | 0.9551 | 1.0000 | 0.8756 | 0.7047 |

**Table A3.** *Cont.*

| Year | State | State | Planting Industry | Forest Industry | Animal Husbandry | Agricultural and Sideline Industry | Aquaculture |
|------|-------|-------|-------------------|-----------------|------------------|-----------------------------------|-------------|
| 2020 | 11 Southeast Asian countries | China | 0.3640 | 0.6170 | 0.5673 | 0.8528 | 0.7684 |
| | | 5 Central Asian countries | 0.6052 | 1.0000 | 0.7492 | 0.7933 | 0.9155 |
| | | Mongolia, Russia | 0.5016 | 0.5340 | 0.5403 | 0.6882 | 0.6871 |
| | | 8 South Asian countries | 0.5044 | 0.9987 | 0.5008 | 0.9067 | 0.7381 |
| | | 19 Central and Eastern European countries | 0.5376 | 0.6201 | 0.5185 | 0.9890 | 0.5560 |
| | | Southwest Asia, 19 Middle Eastern countries | 0.5111 | 0.9747 | 0.5000 | 0.8302 | 0.5937 |
| 2012 | 8 South Asian countries | China | 0.4019 | 0.4727 | 0.5151 | 0.7451 | 0.5847 |
| | | 5 Central Asian countries | 0.3797 | 0.4882 | 1.0000 | 0.9998 | 0.5760 |
| | | Mongolia, Russia | 0.5016 | 0.8506 | 0.5812 | 0.7284 | 0.5445 |
| | | 11 Southeast Asian countries | 0.5018 | 0.7145 | 0.7383 | 0.7689 | 0.9997 |
| | | 19 Central and Eastern European countries | 0.4727 | 0.5672 | 0.6111 | 0.6444 | 0.5789 |
| | | Southwest Asia, 19 Middle Eastern countries | 0.7197 | 0.5657 | 0.6568 | 0.7000 | 0.5964 |
| 2020 | 8 South Asian countries | China | 0.2872 | 0.5205 | 0.7675 | 0.8329 | 0.9056 |
| | | 5 Central Asian countries | 0.5181 | 1.0000 | 0.6892 | 0.7063 | 0.4882 |
| | | Mongolia, Russia | 0.5007 | 0.6055 | 0.7195 | 0.5224 | 0.7527 |
| | | 11 Southeast Asian countries | 0.6074 | 0.6818 | 0.5047 | — | 0.9190 |
| | | 19 Central and Eastern European countries | 0.3788 | 0.5894 | 0.4985 | 0.8287 | 0.4933 |
| | | Southwest Asia, 19 Middle Eastern countries | 0.6085 | 0.5289 | 1.0000 | 0.6105 | 0.5142 |
| 2012 | 19 Central and Eastern European countries | China | 0.5774 | 0.6617 | 0.6038 | 0.8746 | 0.7157 |
| | | 5 Central Asian countries | 0.3448 | 0.5638 | 0.6222 | 0.7972 | 0.7411 |
| | | Mongolia, Russia | 0.5439 | 0.7716 | 0.6463 | 0.9994 | 0.6127 |
| | | 11 Southeast Asian countries | 1.0000 | 0.5060 | 0.5079 | 0.8483 | 0.6181 |
| | | 8 South Asian countries | 1.0000 | 0.6552 | 0.5602 | 0.8087 | 0.7213 |
| | | Southwest Asia, 19 Middle Eastern countries | 0.7165 | 0.7427 | 0.7954 | 0.6637 | 0.6557 |
| 2020 | 19 Central and Eastern European countries | China | 0.3820 | 0.7652 | 0.6055 | 0.8227 | 0.4886 |
| | | 5 Central Asian countries | 0.5171 | 1.0000 | 0.7664 | 0.8687 | 0.6668 |
| | | Mongolia, Russia | 0.9490 | 0.5179 | 0.5832 | 0.8120 | 0.5958 |
| | | 11 Southeast Asian countries | 0.5048 | 0.9659 | 0.5059 | 0.7472 | 0.4992 |
| | | 8 South Asian countries | 0.9999 | 0.5268 | 0.5287 | 0.7127 | 0.9932 |
| | | Southwest Asia, 19 Middle Eastern countries | 0.5258 | 0.5415 | 0.6443 | 0.5633 | 0.5010 |

**Table A3.** *Cont.*

| Year | State | State | Planting Industry | Forest Industry | Animal Husbandry | Agricultural and Sideline Industry | Aquaculture |
|---|---|---|---|---|---|---|---|
| 2012 | Southwest Asia, 19 Middle Eastern countries | China | 0.4218 | 0.6417 | 0.5138 | 0.8692 | 0.7076 |
| | | 5 Central Asian countries | 0.5780 | 0.6078 | 0.9996 | 0.8260 | 0.6781 |
| | | Mongolia, Russia | 0.5244 | 0.9997 | 0.4998 | 0.9814 | 0.5535 |
| | | 11 Southeast Asian countries | 0.5105 | 0.9731 | 0.5077 | 0.8463 | 0.8600 |
| | | 8 South Asian countries | 1.0000 | 0.9993 | 0.7407 | 0.8460 | 0.8958 |
| | | 19 Central and Eastern European countries | 0.8004 | 0.8331 | 0.6824 | 0.8582 | 0.5406 |
| 2020 | Southwest Asia, 19 Middle Eastern countries | China | 0.2762 | 0.5380 | 0.6042 | 0.8909 | 0.6005 |
| | | 5 Central Asian countries | 0.5127 | 0.6908 | 0.7006 | 0.9194 | 0.6100 |
| | | Mongolia, Russia | 0.6576 | 0.5649 | 0.9675 | 0.8577 | 0.9598 |
| | | 11 Southeast Asian countries | 0.6046 | 0.6866 | 0.5198 | 0.8027 | 0.7489 |
| | | 8 South Asian countries | 0.5744 | 0.5299 | 0.8145 | 0.5917 | 0.5098 |
| | | 19 Central and Eastern European countries | 0.6171 | 0.7813 | 0.6252 | 0.8790 | 0.5636 |

**Table A4.** Country Code.

| Number | Country | Number | Country | Number | Country |
|---|---|---|---|---|---|
| 1 | China | 23 | Afghanistan | 45 | Belarus |
| 2 | Kazakhstan | 24 | Nepal | 46 | Moldova |
| 3 | Kyrgyzstan | 25 | Bhutan | 47 | Turkey |
| 4 | Tajikistan | 26 | Sri Lanka | 48 | Iran |
| 5 | Uzbekistan | 27 | Maldives | 49 | Syria |
| 6 | Turkmenistan | 28 | Poland | 50 | Iraq |
| 7 | Mongolia | 29 | Czech Republic | 51 | United Arab Emirates |
| 8 | Russia | 30 | Slovakia | 52 | Saudi Arabia |
| 9 | Vietnam | 31 | Hungary | 53 | Qatar |
| 10 | Laos | 32 | Slovenia | 54 | Bahrain |
| 11 | Cambodia | 33 | Croatia | 55 | Kuwait |
| 12 | Thailand | 34 | Romania | 56 | Lebanon |
| 13 | Malaysia | 35 | Bulgaria | 57 | Oman |
| 14 | Singapore | 36 | Serbia | 58 | Yemen |
| 15 | Indonesia | 37 | Montenegro | 59 | Jordan |
| 16 | Brunei | 38 | Macedonia | 60 | Israel |
| 17 | Philippines | 39 | Bosnia and Herzegovina | 61 | Palestine |
| 18 | Myanmar | 40 | Albania | 62 | Armenia |
| 19 | Timor-Leste | 41 | Estonia | 63 | Georgia |
| 20 | India | 42 | Lithuania | 64 | Azerbaijan |
| 21 | Pakistan | 43 | Latvia | 65 | Egypt |
| 22 | Bangladesh | 44 | Ukraine | | |

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
