# Peer review of "Analysis on Trade Competition and Complementarity of High-Quality Agricultural Products in Countries along the Belt and Road Initiative"

_sustainability, doi:10.3390/su15086671_

Round 1
Reviewer 1 Report
The research deals with analyzing the factors that condition the
similarity and complementarity of the export of agricultural products
and at the same time evaluating the impact on the quality of agricultural
products of countries adhering to the Belt and Road Initiative. The objective
of the investigation is clearly defined, but the article could be better structured in order to make its reading more fluid
and more adherent to the scientific framework that is suitable for such a work.
Here some main considerations:
1) The introduction is well structured and referenced.
2) Materials and Methods. The materials and methods should be better
structured: a) a descriptive premise on the methodological process followed
in the whole work is totally lacking. I would suggest placing this part at the beginning of Materials and Methods.
2) Furthermore, following aspetcs are missing: the description of the methodology related to factors affectingis competitiveness and complementarity (described together with the results
in section 3.2) and the expanded analysis (described together with the
results in section 3.3).
3) Results are well presented but need to be separated from the methods.
4) Discussion, intended as a comparison of the results with other scientific
works is missing.
Minor problems:
1) Line numbers on the manuscript could help review work to suggest the indication of the parts to be moved or changed.
2) Check repetition of a phrase at the end of 2nd page.
3) a map with the 7 zones could help
Author Response
Dear Editor and Reviewer:
Thank you for giving us the opportunity to submit a revised copy of the manuscript. We appreciate the time and effort that you have dedicated to providing your valuable feedback on this manuscript. Your comments and concerns are highly insightful and enabled us to improve the quality of the manuscript. All revisions to the manuscript have been marked up by using tracking changes function. We hope that this revised version has satisfactorily addressed all of your concerns. In the following, the point-by-point responses to each of the comments are presented.
Please find attached for more details.
Sincerely,
Xiao Wang

Reviewer 2 Report
The manuscript entitled "Analysis on Trade Competition and Complementarity of High Quality Agricultural Products in Countries Along the Belt and Road Initiative" (sustainability-2298637) was aimed at studying the factors affecting similarity and trade complementarity of agricultural exports of the main participating countries along the Belt and Road.
To improve the quality of the manuscript, I propose the following:
(1) The abstract should be adapted according to the journal's recommendation, especially regarding this point – the beginning of the abstract should include information on the background: a broad and brief presentation of the general context of the analyzed issue (which is now lacking), as well as a highlight on the literature gap and on the manuscript novelty factor.
(2) I suggest more reflexivity regarding the indicators used to assess competitiveness. For example, the authors embrace the revealed comparative advantage as an essential proxy for international competitiveness, but its limitations should be acknowledged in the manuscript. Please allow me provide a reference to consider in this regard: Istudor, N., Constantin, M., Ignat, R., Chiripuci, B.-C., & Petrescu, I.-E. (2022). The Complexity of Agricultural Competitiveness: Going Beyond the Balassa Index. Journal of Competitiveness, 14(4), 61–77.
(3) I suggest moving Tables 3-5 to the Appendix of the manuscript and elaborate new tables with the descriptive statistics of the RCA, export similarity and trade complementarity index from Tables 3-5. The descriptive statistics should be discussed. Moreover, I believe some visual representations would be more pleasing to the readers instead of the current tabular data presentation.
(4) Before the discussion section, it would add more value to the manuscript if the authors could include a cluster analysis based on RCA, export similarity and trade complementarity index.
(5) The discussion section should be further developed with more managerial implications based on current research findings. The future research avenues proposed by the authors are worthy of investigation, but the current research limitations are not explicitly stated and they should be. Additionally, in the same section, the authors should compare their findings with those of other authors that have published papers in the same field.
Author Response

(The authors gave the same response as above.)

Reviewer 3 Report
I believe that the conclusions can be expanded with more critical considerations highlighting the limits of the model used in the analysis
Author Response

(The authors gave the same response as above.)

Round 2
Reviewer 1 Report
The authors have adequately addressed all the issues raised in the review requests.